# Causal Inference on Distributional Outcomes under Continuous Treatments

## Abstract

Causal inference is widely practiced in various domains. Existing literature predominantly focuses on causal estimators for scalar or vector outcomes. However, real-world scenarios often involve response variables that are better represented as distributions. This paper addresses the need for causal inference methods capable of accommodating the distributional nature of responses when the treatments are continuous variables. We adopt a novel framework for causal inference within a vector space that incorporates the Wasserstein metric. Drawing upon Rubin's causal framework, we introduce three estimators, namely the Distributional Direct Regression (Dist-DR), Distributional Inverse Propensity Weighting (Dist-IPW), and Distributional Doubly Machine Learning (Dist-DML) estimators, tailored for estimating target quantities, i.e., causal effect maps. We thoroughly examine the statistical properties of these estimators. Through two experiments, we validate the efficacy of the proposed methodology, establishing its practical utility.

## 1 Introduction

The investigation of how treatments influence outcomes, known as causal inference, is a common practice across diverse domains, e.g., medical (Robins et al., 2000) and finance Huang et al. (2021). To explore these effects, researchers have introduced and studied different causal estimators, such as the average treatment effect (ATE), the quantile treatment effect (QTE), and the conditional average treatment effect (CATE) (Chernozhukov & Hansen, 2005; Chernozhukov et al., 2018; Abrevaya et al., 2015; Hartman et al., 2015).

However, all the aforementioned causal quantities that appear in the literature primarily center on scenarios where *the realization of the outcome variable for each unit can be represented as a scalar or vector*. However, there are many practical situations where the response for each unit should be described as a *distribution*. An illustrative example can be found in the investigation of the impact of working hours on individuals' activity intensity behaviors. One's activity intensities are typically recorded at regular intervals (e.g., 1 min), and these records collectively form an activity distribution that encapsulates an individual's activity behavior. Notably, different users may exhibit various activity distributions. For instance, as depicted in Figure 1a, the activity intensity distributions of 10 users are displayed, each exhibiting distinct preferences for various activity intensities.

Moreover, consider the scenario in Figure 1b, where two users (A and B) initially have the same activity intensity distribution with a mean of 30. Upon adopting treatments, User A increases intensity for all activities by 20 units, resulting in a rightward shift of the distribution by 20 units, while the shape remains unchanged. Consequently, the mean of the distribution increases from 30 to 50. On the other hand, User B only enhances intensity for high-intensity activities, leading to a significant transformation in the distribution's shape. Nonetheless, the distribution's mean remains at 50. In this context, focusing solely on scalar outcomes as causal quantities in literature, e.g., the mean of the activity intensity distribution, fails to reveal the distinct behavioral patterns of these two users.

As such, there arises a need for causal inference methods that can account for the distributional nature of responses, enabling a more accurate characterization of treatment effects. This paper endeavors to fill this gap by exploring causal inference within a vector space encompassing a spectrum of distributions in scenarios featuring continuous treatment variables. We first equip such vector space with a suitable metric for quantifying dissimilarity between distributions. In contrast to the conventional *Euclidean metric*, which merely averages distributions pointwisely, we opt for the

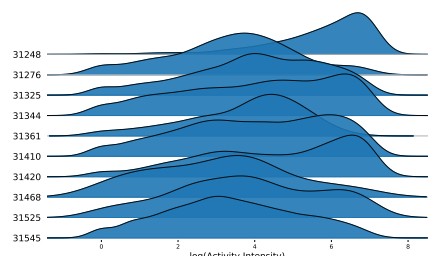

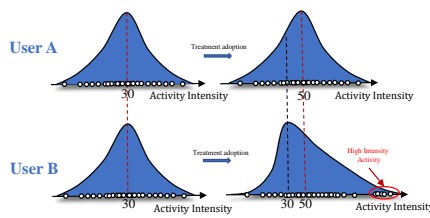

(a) The activity intensity distributions of 10 users from the real-world dataset.

(b) The impact of treatment adoption on activity intensity distribution shift (one point stands for one activity record).

Figure 1: Example of Activity Intensity Distributions.

*Wasserstein metric*, renowned for preserving the inherent structure of random distributions more effectively.

Grounded in Rubin's foundational causal framework, we introduce three distinct estimators for target quantities, termed the *causal effect map*, which is analogous to the ATE in the classical causal framework. We comprehensively explore the statistical asymptotic properties that underlie these estimators. Subsequently, to empirically ascertain the efficacy of our proposed methodologies, we conduct two experiments, including one simulation and one real-world dataset. Our findings underscore the effectiveness of all three estimators.

The contributions of this paper are threefold:

- We introduce a novel non-parametric framework and three distinct cross-fitted estimators for inferring causal effects when the treatment variable takes continuous values.

- We study the asymptotic properties characterizing the cross-fitted estimators, offering valuable insights into the statistical performance and reliability of the proposed estimators.

- We perform two experiments to validate our proposed estimator, and the results from the numerical experiments are consistent with our theoretical findings.

## 2 RELATED WORK

The key assumption of classical causal inference is that, given the treatment $A = a$, the realization of response variables for each unit is a *scalar point* drawing from the same potential outcome distribution. Under the assumption, several causal quantities are introduced and studied. For instance, ATE (Chernozhukov et al., 2018) is the difference between the means of any two potential outcome distributions (i.e., $\mathbb{E}[Y(A = \bar{a})] - \mathbb{E}[Y(A = a)]$). CATE is the mean difference of two potential outcomes in the total population conditioning on some covariates (Fan et al., 2022). Instead of studying the mean of potential outcome distribution, QTE (Chernozhukov & Hansen, 2005) focus on the difference between two potential outcome distributions at $\tau$-quantiles (i.e., $Q(\tau, Y(A = \bar{a})) - Q(\tau, Y(A = a))$).

The general approach to estimating the causal effect between treatment and outcome is constructing the estimators for the target quantities. The simplest method, called the Direct Regression (DR) approach, is regressing the relation between the response and the features pair of treatment and covariates. However, the estimated relation from the observed dataset can be biased since the dataset is always not randomized. To address the issues, the inverse propensity weighting (IPW) method is introduced (Rosenbaum & Rubin, 1983; Hirano et al., 2003), aiming to formulate a pseudo-population and obtain the estimators for the target quantities in the pseudo-population. However, using the estimated extreme propensity score always gives the estimates with large variance. To overcome the challenges, the Doubly Machine Learning (DML) approach is proposed, which is endowed with a *doubly robust property* (Chernozhukov et al., 2018), (Colangelo & Lee, 2019).

The above methods are restricted when the outcome of each unit includes many observations or points, and they constitute a distribution. Thus, it is necessary to seek alternative frameworks for

distributional outcomes. Indeed, the distribution can be treated as a special case of functional outcome and is closely related to the field of functional data analysis (FDA) that analyzes data under information varying over a continuum (Cai et al., 2022; Chen et al., 2016). Specifically, Ecker et al. (2023) considers a causal framework to study the impact of treatment on the functional outcome. However, their work conducts causal inference in Euclidean space, in which the random structure of the distributional outcome can be destroyed (Verdinelli & Wasserman, 2019; Panaretos & Zemel, 2019). As such, Lin et al. (2021) considers the causal study in the Wasserstein space, but they only consider the treatment variable taking binary values. We consider extending their framework to continuous treatment and propose three distinct estimators. We provide more detailed comparisons between our framework and classical framework in Appendix B.

## 3 BACKGROUND

### 3.1 NOTATIONS

In this paper, we adopt the notation $A \in \mathcal{A} \subset \mathbb{R}$ to denote the *continuous treatment variable*. The $m$-dimensional vector $\mathbf{X} = [X^1, \cdots, X^m] \in \mathcal{X}$ corresponds to the *covariates/confoundings*. The *response variable* is denoted as $\mathcal{Y}$, and we use $\mathcal{Y}(a)$ to signify the response variable associated with a specific value $a$. We assume that the realization of $\mathcal{Y}$ and $\mathcal{Y}(a)$ is a distribution instead of a scalar value. Specifically, we focus on a sub-case where the functional response corresponds to the cumulative distribution function (CDF) within the Wasserstein space denoted as $\mathcal{W}_2(\mathcal{I})$. We finally assume that there exist $N$ samples denoted as $(\mathbf{X}_s, A_s, \mathcal{Y}_s)_{s=1}^N$.

### 3.2 CAUSAL ASSUMPTIONS

As with the previous studies Rubin (1978; 2005), our approach relies on four assumptions. They are (1) *Stable Unit Treatment Unit Assumption*, (2) *Consistency* , *Ignorability* , and (4) *Overlap* . We defer detailed assumptions about these assumptions in Appendix A.

### 3.3 WASSERSTEIN SPACE

Given that the outcome in our paper is selected as the CDF, it becomes crucial to define a vector space that encompasses the CDF and establish an appropriate distance measure to compare and contrast two CDFs effectively. To begin, let $\mathcal{I} \subset \mathbb{R}$, we define the vector space $\mathcal{W}p(\mathcal{I})$ that comprises CDFs defined on $\mathcal{I}$ and satisfying the condition:

$$\mathcal{W}_p(\mathcal{I}) = \left\{ \lambda \text{ is a CDF on } \mathcal{I} \mid \int_{\mathcal{I}} t^p d\lambda(t) < \infty \right\}, \quad \text{where } p \geq 1.$$

Using Jensen's inequality, we can conclude that $\left( \int_{\mathcal{I}} t^q d\lambda(t) \right)^{\frac{1}{q}} \leq \left( \int_{\mathcal{I}} t^p d\lambda(t) \right)^{\frac{1}{p}}$ when $1 \leq q \leq p$. Hence, $\mathcal{W}_p(\mathcal{I})$ contains all the CDF $\lambda$ with all the finite order central moment up to $p$-th order.

We then establish a distance metric between two CDFs. The simplest measure that can be employed is the *Euclidean p-measure*, where the distance between two CDFs can be computed as the differences between the two CDFs point-wisely. Mathematically, given two CDFs $\lambda_1$ and $\lambda_2$ defined on $\mathcal{I}$, the Euclidean $p$-measure is $(\int_{\mathcal{I}} |\lambda_1(t) - \lambda_2(t)|^p dt)^{\frac{1}{p}}$.

However, the Euclidean $p$-measure is not an optimal metric for characterizing the distance between two CDFs since averaging all the values of the distributions will destroy the structural properties of the resulting distribution, leading to a loss of essential characteristics. A concrete illustration of this issue is provided in Figure 2, which showcases ten distributions with distinct means and variances in the top plot. When these distributions are averaged using the Euclidean metric, the resulting green line in the bottom plot demonstrates that the bell shape characteristic of a normal distribution is not preserved.

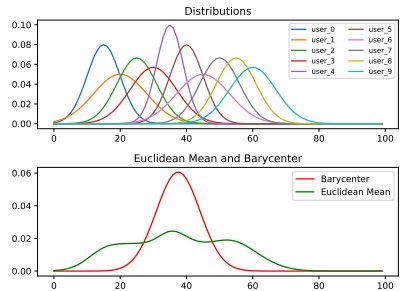

Figure 2: Examples for the average distribution of the 10 distributions using the Euclidean and Wasserstein metric.

Apart from the usual Euclidean measure, we can also use the $p$-Wasserstein metric (Villani, 2021; Panaretos & Zemel, 2019; Feyeux et al., 2018), which is defined as

**Definition 1** *Given two random variables $V_1$ and $V_2$, let the marginal CDFs of $V_1$ and $V_2$ be $\lambda_1$ and $\lambda_2$ that are defined on $\mathcal{I}$. Besides, let $\Lambda$ be the set containing all the joint densities of $V_1$ and $V_2$. The $p$-Wasserstein metric is given as $\mathbb{D}_p(\lambda_1, \lambda_2)$ such that*

$$\mathbb{D}_p(\lambda_1, \lambda_2) = \left\{ \inf_{\tilde{\lambda} \in \Lambda} \int_{\mathcal{I} \times \mathcal{I}} \gamma(s,t)^p d\tilde{\lambda}(s,t) \right\}^{\frac{1}{p}}. \tag{1}$$

In Definition 1, $\gamma(s,t)$ is a function such that $\gamma(s,t) : \mathbb{R} \times \mathbb{R} \to \mathbb{R}$ satisfies the metric axioms: *positivity axiom*, *symmetry axiom*, and *triangle inequality axiom*. Moreover, $\gamma(\cdot, \cdot)$ represents the *cost* of transporting a point mass located at $s$ following the distribution $\lambda_1$ to $t$ following the distribution $\lambda_2$. As a result, the integral $\int_{\mathcal{I} \times \mathcal{I}} \gamma(s,t)^p d\tilde{\lambda}(s,t)$ represents the *total cost* of transporting points from $\lambda_1$ to $\lambda_2$ given that the joint distribution of $\lambda_1$ and $\lambda_2$ is $\tilde{\lambda}$. $\mathbb{D}_p(\lambda_1, \lambda_2)$ is thus the minimum cost among all joint distributions of $(\lambda_1, \lambda_2)$.

The vector space $\mathcal{W}_p(\mathcal{I})$ equipped with the metric $\mathbb{D}_p(\cdot, \cdot)$ forms the *p-Wasserstein space* (denoted as $(\mathcal{W}_p(\mathcal{I}), \mathbb{D}_p(\cdot, \cdot))$). Since the function $\gamma(s,t)$ in Definition 1 satisfies the metric axioms, the distance measures $\mathbb{D}_p(\cdot, \cdot)$ also satisfies the metric axioms. Consequently, the $p$-Wasserstein space is indeed a metric space. In the sequel, we assume that $p = 2$ and $\gamma(s,t) = |s - t|$.

One of the significant advantages of using the Wasserstein metric is its ability to preserve the structural properties of the distributions being averaged. As in Figure 2, the red line represents the average of all ten normal distributions computed using the Wasserstein metric, and it retains the shape of normal distributions.

## 4 CAUSAL QUANTITIES

### 4.1 CAUSAL MAP AND CAUSAL EFFECT MAP

Similar to the ATE, the target quantity in our paper is called *causal effect map*, which provides a comprehensive understanding of the treatment-response relationships.

**Definition 2** *The causal effect map $\triangle_{a\bar{a}}$ between treatments $a$ and $\bar{a}$ is defined as*

$$\triangle_{a\bar{a}} := \triangle_a - \triangle_{\bar{a}} := \mu_a^{-1} - \mu_{\bar{a}}^{-1}, \tag{2}$$

*where $\mu_a := \underset{v \in \mathcal{W}_2(\mathcal{I})}{\arg\min} \mathbb{E}\big[\mathbb{D}_2(\mathcal{Y}(a), v)^2\big]$. We also term $\triangle_a$ as the casual map of treatment $a$.*

Here, the realization of $\mathcal{Y}(a)$ is a distribution. The quantity $\mathbb{E}\big[\mathbb{D}_2(\mathcal{Y}(a), v)\big]$ represents the average Wasserstein distance centered at $v \in \mathcal{W}_2(\mathcal{I})$. As a result, the average Wasserstein distance centered at $\mu_a$ is the smallest, and it is commonly referred to as the *Wasserstein barycenter*. Notably, $\mu_a$ is a CDF, and thus $\mu_a^{-1}$ is the inverse function of CDF, which is also known as the *quantile function*.

### 4.2 PROPERTIES OF CAUSAL MAP/CAUSAL EFFECT MAP

From the previous section, we have shown that $\triangle_a = \mu_a^{-1}$ where $\mu_a := \underset{v \in \mathcal{W}_2(\mathcal{I})}{\arg\min} \mathbb{E}\big[\mathbb{D}_2(\mathcal{Y}(a), v)^2\big]$.

The calculation $\triangle_a(\cdot)$ requires solving an optimization problem in the Wasserstein space. This optimization step can be computationally demanding, particularly when dealing with high-dimensional data or large sample sizes. To enhance the efficiency, we simplify the calculation of $\triangle_a(\cdot)$ and eliminate the optimization step. We conclude this point in Proposition 1:

**Proposition 1** *Given that Assumptions 1 - 4 hold, we have $\triangle_a = \mathbb{E}\big[\mathcal{Y}(a)^{-1}\big]$.*

We defer the proof in Appendix C. $\mathbb{E}[\mathcal{Y}^{-1}(a)]$ represents the expectation of the inverse CDF when all units in the population receive treatment $a$.

### 4.3 ESTIMATORS

In practice, we often encounter situations where not all individuals receive treatment $a$, and in some cases, no individuals receive treatment $a$, especially when $A$ is a continuous variable. To address this challenge and facilitate practical estimations from observed datasets, we further explore three alternative estimators of $\mathbb{E}[\mathcal{Y}^{-1}(a)]$, namely **Dist**ributional **D**irect **R**egression (**Dist-DR**) estimator, **Dist**ributional **I**nverse **P**ropensity **W**eighting (**Dist-IPW**) estimator, and **Dist**ributional **D**oubly **M**achine **L**earning (**Dist-DML**) estimator.

**Dist-DR estimator** can be obtained simply using Causal Assumptions (2) - (3). Indeed, we have

$$\triangle_a = \mathbb{E}[\mathcal{Y}(a)^{-1}] = \mathbb{E}[\mathbb{E}[\mathcal{Y}(a)^{-1}|\mathbf{X}]] \overset{*}{=} \mathbb{E}[\mathbb{E}[\mathcal{Y}(a)^{-1}|A=a,\mathbf{X}]] \overset{\star}{=} \mathbb{E}[\mathbb{E}[\mathcal{Y}^{-1}|A=a,\mathbf{X}]]. \quad (3)$$

Here, $\star$ follows from Causal Assumption (2) while $*$ follows from Causal Assumption (3). Let us define $m_a(\mathbf{X}) = \mathbb{E}[\mathcal{Y}^{-1}|A=a,\mathbf{X}]$ which is a regression function that can be estimated using any functional regression methods, e.g., Chen et al. (2016). As such, we obtain the Dist-DR estimator $\triangle_{a;DR}$ using sample averaging such that

$$\triangle_{a;DR} = \frac{1}{N}\sum_{s=1}^{N} m_a(\mathbf{X}_s). \quad (4)$$

However, the Dist-DR estimator neglects the potential influence of the covariates $\mathbf{X}$ on the treatment variable $A$ and is not suitable to construct estimators for causal analysis unless the observed dataset is random. Thus, we consider to express $\mathbb{E}[\mathcal{Y}(a)^{-1}]$ as other forms.

**Dist-IPW estimator** uses the Horvitz–Thompson Theorem (Horvitz & Thompson, 1952; Overton & Stehman, 1995), and we can show that

**Proposition 2** *Given that Assumptions 1 - 4 hold, we have*

$$\triangle_a = \mathbb{E}\left[\frac{\delta(A-a)}{p(a|\mathbf{X})}\mathcal{Y}^{-1}\right]. \quad (5)$$

*Here, $\delta(\cdot)$ is known as the Delta Dirac function.*

In Eqn. 5, the term $\frac{\delta(A-a)}{p(a|\mathbf{X})}$ serves as the weight to construct a pseudo-population, where groups with a smaller portion in the dataset receive larger weights, while groups with a larger portion receive smaller weights. These weights are usually constructed using the (generalized) propensity scores, which capture the likelihood of receiving treatment based on covariates.

We defer the proof in Appendix D. Unlike the Dist-DR estimator, we cannot directly construct estimators according to equation 5 using sample averaging due to the presence of the Delta Dirac function $\delta(A-a)$. To overcome this, we usually replace the Delta Dirac function with some *Kernel Approximations*.

**Definition 3 (Kernel function)**

1. *Given that $K(\cdot) : \mathbb{R} \to \mathbb{R}$ is a symmetric function (i.e.,$K(v) = K(-v) \ \forall v \in \mathbb{R}$). We say that $K(\cdot)$ is a kernel function if it satisfies $\int_{\mathbb{R}} K(v)dv = 1$.*

2. *A kernel function $K(\cdot)$ is said to have order $\nu$ ($\nu$ is an even number) if $\int_{\mathbb{R}} v^j K(v) \ dv = 0$ $\forall \ 1 \le j \le \nu - 1$ and $\int_{\mathbb{R}} v^\nu K(v) \ dv < \infty$.*

In this paper, we concentrate on the second-order kernel function and present some commonly used second-order kernels in Appendix D. For any arbitrary kernel function $K(x)$, we can define the *scaled kernel* with *bandwidth* $h$. It is denoted as $K_h(x)$ such that $K_h(x) := \frac{1}{h}K(\frac{x}{h})$. Due to the fact that $\lim_{h \to 0} K_h(x) = \delta(x)$, we can replace $\delta(A-a)$ in equation 5 with $K_h(A-a)$, and we can then construct the Dist-IPW estimator $\triangle_{a;IPW}$ using sample averaging such that

$$\triangle_{a;IPW} = \frac{1}{N}\sum_{s=1}^{N} \frac{K_h(A_s-a)}{p(a|\mathbf{X}_s)}\mathcal{Y}_s^{-1}. \quad (6)$$

The Dist-DR estimator uses the nuisance parameter $\mathbb{E}[\mathcal{Y}^{-1}|A = a, \mathbf{X}]$ only, while the Dist-IPW estimator uses the nuisance parameter $p(a|\mathbf{X})$ only. Naturally, we can derive an estimator that requires both the nuisance parameters $\mathbb{E}[\mathcal{Y}^{-1}|A = a, \mathbf{X}]$ and $p(a|\mathbf{X})$.

**Dist-DML estimator** is indeed developed from the Doubly Machine Learning Theorem as depicted in Chernozhukov et al. (2018). The theorem provides a powerful framework that combines the benefits of both the Dist-DR estimator and the Dist-IPW estimator. To start with, we show that $\triangle_a$ can be expressed in terms of $\mathbb{E}[\mathcal{Y}^{-1}|A = a, \mathbf{X}]$ and $p(a|\mathbf{X})$ in Proposition 3.

**Proposition 3** *Denote* $m_a(\mathbf{X}) = \mathbb{E}[\mathcal{Y}^{-1}|A = a, \mathbf{X}]$. *Suppose that Assumptions 1 - 4 hold, we have*

$$\triangle_a = \mathbb{E}\left[m_a(\mathbf{X}) + \frac{\delta_a(A)}{p(a|\mathbf{X})}[\mathcal{Y}^{-1} - m_a(\mathbf{X})]\right]. \tag{7}$$

We defer the proof in Appendix E. Moreover, as the Dist-DR and Dist-IPW estimators, we can also estimate the Dist-DML estimator $\triangle_{a;DML}$ using sample averaging such that

$$\triangle_{a;DML} = \frac{1}{N}\sum_{s=1}^{N}\left[m_a(\mathbf{X}_s) + \frac{K_h(A_s - a)}{p(a|\mathbf{X}_s)}(\mathcal{Y}_s^{-1} - m_a(\mathbf{X}_s))\right]. \tag{8}$$

The Dist-DML estimator possesses a valuable property known as *doubly robustness*, where equation 7 still hold even if either $p(a|\mathbf{X})$ or $m_a(\mathbf{X})$, but not both, are misspecified. We prove this property in Appendix F. Further, the estimation accuracy of $m_a(\cdot)$ and $p(a|\mathbf{X})$ can be reduced if the Dist-DML estimator is used in lieu of the Dist-DR estimator and the Dist-IPW estimator (see Theorem 2 in Appendix H).

## 4.4 ALGORITHM

In the previous section, we have derived the estimators $\triangle_{a;DR}$, $\triangle_{a;IPW}$, and $\triangle_{a;DML}$. In order to obtain estimations of these estimators based on an observed dataset, we employ the cross-fitting technique, which can help mitigate the risk of over-fitting (Chernozhukov et al., 2018).

In particular, we partition the $N$ samples into $\mathcal{K}$ disjoint groups, where the $k^{\text{th}}$ group is denoted as $\mathcal{D}_k$ and contains $N_k$ samples, for all $k = \{1, \ldots, \mathcal{K}\}$. Let $\mathcal{D}_{-k} = \cup_{r=1,r\neq k}^{\mathcal{K}}\mathcal{D}_r$, and we use $\mathcal{D}_{-k}$ to learn the estimated nuisance parameters $\hat{m}_a^k(\mathbf{X})$ and $\hat{p}^k(a|\mathbf{X})$ of $m_a(\cdot)$ and $p(a|\cdot)$. We also suppose that the empirical estimation of $\mathcal{Y}$ is denoted as $\hat{\mathcal{Y}}$. Subsequently, we utilize $\mathcal{D}_k$ to compute

$$\hat{\triangle}_{a;DR}^k = \frac{1}{N_k}\sum_{s\in\mathcal{D}_k}\hat{m}_a^k(\mathbf{X}_s), \quad (9) \qquad \hat{\triangle}_{a;IPW}^k = \frac{1}{N_k}\sum_{s\in\mathcal{D}_k}\frac{K_h(A_s - a)}{\hat{p}^k(a|\mathbf{X}_s)}\hat{\mathcal{Y}}_s^{-1}, \quad (10)$$

$$\hat{\triangle}_{a;DML}^k = \frac{1}{N_k}\sum_{s\in\mathcal{D}_k}\left[\hat{m}_a^k(\mathbf{X}_s) + \frac{K_h(A_s - a)}{\hat{p}^k(a|\mathbf{X}_s)}(\hat{\mathcal{Y}}_s^{-1} - \hat{m}_a^k(\mathbf{X}_s))\right]. \tag{11}$$

Consequently, we can obtain the cross-fitted estimators $\hat{\triangle}_{a;w}$ such that

$$\hat{\triangle}_{a;w} = \sum_{k=1}^{\mathcal{K}}\frac{N_k}{N}\hat{\triangle}_{a;w}^k, \tag{12}$$

where $w \in \{\text{Dist-DR}, \text{Dist-IPW}, \text{Dist-DML}\}$. We finally present an Algorithm that summarizes the procedures of getting the estimates of the cross-fitted estimators $\hat{\triangle}_{a;w}$ in Appendix G.

## 5 THEORY

In this section, we aim to study the asymptotic properties of the proposed estimator $\hat{\triangle}_{a;w}$ for any $w \in \{\text{Dist-DR}, \text{Dist-IPW}, \text{Dist-DML}\}$. Let $\mathbf{X}$ be a random variable with distribution $F_{\mathbf{X}}(\mathbf{x})$. Generally, we consider three types of $\mathcal{L}^2$ space containing different forms of function: i) $f : \mathcal{X} \rightarrow \mathbb{R}$; ii) $g, \tilde{g} : [0, 1] \rightarrow \mathbb{R}$; and iii) $\Gamma : \mathcal{X} \times [0, 1] \rightarrow \mathbb{R}$. For the second type of $\mathcal{L}^2$ space, we can define an

inner product $\langle \cdot, \cdot \rangle$ such that $\langle g, \tilde{g} \rangle = \int_{[0,1]} g(t)\tilde{g}(t)dt$ where $\int_{[0,1]} |g(t)|^2 dt, \int_{[0,1]} |\tilde{g}(t)|^2 dt < \infty$. For each $\mathcal{L}^2$ space, we have the corresponding norm: i) $f(\mathbf{X})$ as $\|f(\mathbf{X})\|_2^2 = \int_{\mathcal{X}} |f(\mathbf{x})|^2 dF_{\mathbf{X}}(\mathbf{x}) = \mathbb{E}[|f(\mathbf{X})|^2]$; ii) $\|g\|^2 = \int_{[0,1]} g(t)^2 dt$; and iii) $\|f(\mathbf{X},t)\|^2 = \int_{\mathcal{X}} \|f(\mathbf{x},t)\|^2 dF_{\mathbf{X}}(\mathbf{x})$.

We also let $\mathbb{P}_N$ be the empirical average operator such that $\mathbb{P}_N \mathcal{O} = \frac{1}{N} \sum_{s=1}^N \mathcal{O}_s$. We use $\tilde{m}_a^k(\cdot)$ and $\hat{m}_a^k(\cdot)$ to denote the estimates of $m_a(\cdot)$ using the outcome $Y$ and $\hat{Y}$ based on the set $\mathcal{D}_{-k}$ respectively. Simultaneously, let $\rho_m^4 = \sup\{\|\|\tilde{m}_a^k - m_a\|\|^4, \ a \in \mathcal{A}\} = \sup\{[\int \|\tilde{m}_a^k(\mathbf{x}) - m_a(\mathbf{x})\|^2 dF_{\mathbf{X}}(\mathbf{x})]^2, \ a \in \mathcal{A}\}$ for $1 \le k \le \mathcal{K}$ and define $\rho_p^4 = \sup_{a \in \mathcal{A}} \mathbb{E}[|\hat{p}^k(a|\mathbf{X}) - p(a|\mathbf{X})|^4]$. Finally, we present the convergence assumptions that are required in studying the asymptotic properties of the proposed estimators.

**Convergence Assumption 1** *The estimates $\hat{\mathcal{Y}}_1, \cdots, \hat{\mathcal{Y}}_N$ are independent of each other. Further, there are two sequences of constants $\alpha_N = o(N^{-\frac{1}{2}})$ and $\nu_N = o(N^{-\frac{1}{2}})$ (note that $o(N^{-\frac{1}{2}})$ implies $o(1)$ automatically) such that*

$$\sup_{1 \le s \le N} \sup_{v \in \mathcal{W}(\mathcal{I})} \mathbb{E}[\mathbb{D}_2^2(\hat{\mathcal{Y}}_s, \mathcal{Y}_s)|\mathcal{Y}_s = v] = O(\alpha_N^2) \quad and \quad \sup_{1 \le s \le N} \sup_{v \in \mathcal{W}(\mathcal{I})} \mathbb{V}[\mathbb{D}_2^2(\hat{\mathcal{Y}}_s, \mathcal{Y}_s)|\mathcal{Y}_s = v] = O(\nu_N^4).$$

**Convergence Assumption 2** *$\forall \ a \in \mathcal{A}$ and $\forall \ 1 \le k \le \mathcal{K}$, we have*

1. $\sup_{\mathbf{x} \in \mathcal{X}} \|\tilde{m}_a^k(\mathbf{x}) - m_a(\mathbf{x})\| = o_P(1) \quad and \quad \sup_{\mathbf{x} \in \mathcal{X}} \|\hat{p}^k(a|\mathbf{x}) - p(a|\mathbf{x})\| = o_P(1).$

2. $\|\|\hat{m}_a^k - \tilde{m}_a^k\|\| = O_P(N^{-1} + \alpha_N^2 + \nu_N^2).$

3. *There exist constants $c_1$ and $c_2$ such that $0 < c_1 \le \frac{N_k}{N} \le c_2 < 1$ for all $N$.*

In Theorem 1, we only present the asymptotic properties of $\hat{\triangle}_{a;DML}$. For other cases, we defer the asymptotic studies to the Appendix H.

**Theorem 1** *Suppose that $p(a|\mathbf{x}) \in \mathcal{C}^3$ on $\mathcal{A}$ such that the derivatives (including 0-th order derivative) are bounded uniformly in the sample space for any $\mathbf{x}$. Further, we assume that $\mathbb{E}[\mathcal{Y}^{-1}|A = a, \mathbf{X}] \in \mathcal{C}^3$ on $[0,1] \times \mathcal{A}$ and $\mathbb{E}[\|\mathcal{Y}^{-1}\||A = a, \mathbf{X}] \in \mathcal{C}^3$ on $\mathcal{A}$ which are bounded uniformly in the sample space. If $h \to 0$, $Nh \to \infty$, and $Nh^5 \to C \in [0, \infty)$, then, under the convergence assumptions, we have*

$$\sqrt{Nh}(\hat{\triangle}_{a;w} - \triangle_a) = \sqrt{Nh}\left[\mathbb{P}_N\{\varphi(A, \mathbf{X}, \mathcal{Y})\} - \triangle_a\right] + o_P(1), \tag{13}$$

*where $\varphi(A, \mathbf{X}, \mathcal{Y}) = \frac{K_h(A-a)\{\mathcal{Y}^{-1} - m_a(\mathbf{X})\}}{p(a|\mathbf{X})} + m_a(\mathbf{X})$ if $w = DML$ and $\rho_m \rho_p = o(N^{-\frac{1}{2}})$, $\rho_m = o(1)$, $\rho_p = o(1)$. Additionally, $\sqrt{Nh}\{\hat{\triangle}_{a;w} - \triangle_a - h^2 B_a\}$ converges weakly to a centred Gaussian process in $\mathcal{L}^2([0,1])$ where $B_a = \left(\int u^2 K(u)du\right) \times \left(\mathbb{E}\left[\frac{\partial_a m_a(\mathbf{X})\partial_a p(a|\mathbf{X})}{p(a|\mathbf{X})}\right] + \frac{1}{2}\mathbb{E}[\partial_{aa}^2 m_a(\mathbf{X})]\right).$*

We also defer the proofs of Theorem 1 to the Appendix H. Note that if estimators are constructed from the Dist-DML form, the accuracy in estimating nuisance parameters can be relaxed. We only require that $\rho_m \rho_p$ equals $o(N^{-\frac{1}{2}})$. For example, we can have both $\rho_m$ and $\rho_p$ equal $o(N^{-\frac{1}{4}})$ if the Dist-DML estimator is used, but we must have $\rho_m$ and $\rho_p$ equal $o(N^{-\frac{1}{2}})$ if either the Dist-DR estimator or the Dist-IPW estimator is used (see Appendix H).

## 6 SIMULATION EXPERIMENT

To validate our theoretical results, we conduct a simulated experiment where the treatment variable $A$ takes continuous values. The outcome $\mathcal{Y}_s^{-1}$ for each unit is simulated as

$$\mathcal{Y}_s^{-1}(A_s) = c + (1-c)(\mathbb{E}[A_s] + \exp(A_s)) \times \sum_{j=1}^{\frac{m}{2}} \frac{\exp(X_s^{2j-1} X_s^{2j})}{\sum_{k=1}^{\frac{m}{2}} \exp(X_s^{2k-1} X_s^{2k})} \mathbf{B}^{-1}(\alpha_j, \beta_j) + \epsilon_s. \tag{14a}$$

Table 1: The experiment results for three estimators on treatment $A = 0.00$. The reported values are averages across 100 experiments, with Std. in parentheses. The best results are highlighted in bold.

| | Q=0.1 | Q=0.2 | Q=0.3 | Q=0.4 | Q=0.5 | Q=0.6 | Q=0.7 | Q=0.8 | Q=0.9 | Error |
|---|---|---|---|---|---|---|---|---|---|---|
| **Ground** | 0.0112 | 0.0462 | 0.1083 | 0.2271 | 0.5026 | 0.7782 | 0.8970 | 0.9591 | 0.9941 | |
| **Dist-DR** | 0.0101 | 0.0364 | 0.1412 | 0.3009 | 0.4917 | 0.6879 | 0.8561 | 0.9609 | 0.9670 | |
| | (0.0050) | (0.0027) | (0.0029) | (0.0045) | (0.0064) | (0.0079) | (0.0100) | (0.0124) | (0.0169) | |
| **Dist-DR-MAE** | 0.0011 | 0.0099 | 0.0329 | 0.0738 | 0.0109 | 0.0903 | 0.0409 | 0.0019 | 0.0271 | 0.0321 |
| **Dist-IPW** | 0.0071 | 0.0557 | 0.1240 | 0.2424 | 0.4817 | 0.7064 | 0.8190 | 0.8809 | 0.9293 | |
| | (0.0004) | (0.0014) | (0.0031) | (0.0063) | (0.0129) | (0.0208) | (0.0240) | (0.0257) | (0.0271) | |
| **Dist-IPW-MAE** | 0.0041 | 0.0095 | 0.0158 | 0.0153 | 0.0210 | 0.0718 | 0.0780 | 0.0781 | 0.0648 | 0.0398 |
| **Dist-DML** | 0.0080 | 0.0589 | 0.1353 | 0.2658 | 0.5195 | 0.7591 | 0.8846 | 0.9547 | 1.0039 | |
| | (0.0006) | (0.0010) | (0.0009) | (0.0021) | (0.0034) | (0.0024) | (0.0019) | (0.0021) | (0.0019) | |
| **Dist-DML-MAE** | 0.0032 | 0.0127 | 0.0270 | 0.0387 | 0.0169 | 0.0190 | 0.0124 | 0.0044 | 0.0098 | **0.0160** |

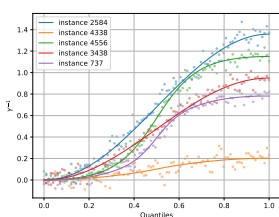
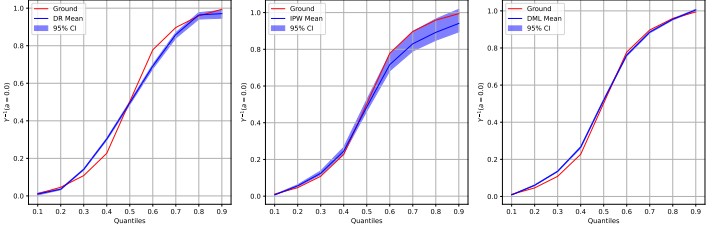

Figure 3: The inverse CDF of 5 simulated units.

Figure 4: The estimated quantile function when $A = 0.00$ from Dist-DR (left), Dist-IPW (middle), and Dist-DML (right) methods.

Here, $m$ is an even number that indicates the number of covariates. $\mathbf{B}^{-1}(\alpha, \beta)$ is the inverse CDF of Beta distribution with the shapes' parameters $\alpha$ and $\beta$. We choose Beta distributions since they vary widely given different parameters. The constant $c$ controls the strength of the causal relationship between $A_s$ and $\mathcal{Y}_s^{-1}$. $\epsilon_s$ is the noise that follows $\mathcal{N}(0, 0.05)$. Then, the treatment $A_s$ for each unit is generated by

$$A_s \sim \mathcal{N}(\gamma^\top \mathbf{X}_s, \log(1 + \exp(\delta^\top \mathbf{X}_s))). \tag{14b}$$

Since the ground truth outcome and the predicted outcome are functions, we thus discretize them and compare the mean absolute error (MAE) between ground truth outcome $\triangle_a$ and estimated causal effect map $\hat{\triangle}_a$ on 9 quantiles with levels ranging from 0.1 to 0.9. We repeat the experiment 100 times to report the mean and standard deviation of MAE.

**Experiment Settings** We choose $m = 10$ such that $X^1, X^2 \sim \mathcal{N}(-2, 1), X^3, X^4 \sim \mathcal{N}(-1, 1), X^5, X^6 \sim \mathcal{N}(0, 1), X^7, X^8 \sim \mathcal{N}(1, 1)$, and $X^9, X^{10} \sim \mathcal{N}(2, 1)$. Within each unit, 100 observations are generated in accordance with equation 14a using the inverse transform sampling technique. In total, 5,000 units are generated. Figure 3 offers a visual representation of 5 simulated instances, showcasing the variability in outcome functions across different units.

We first estimate the functional regression $\hat{m}_a(\mathbf{X}_s)$ by regressing $\hat{\mathcal{Y}}^{-1}$ w.r.t. $(A, \mathbf{X})$. Then, conventional methods might assume a specific form for $p(a|\mathbf{X})$, such as a linear form (Su et al., 2019), or employ kernel-based techniques (Colangelo & Lee, 2019). We adopt a generative approach to estimate the density function, drawing inspiration from Grathwohl et al. (2019).

**Experiment Results** We conduct the experiment across three distinct treatment levels: $A = -0.05$, $A = 0.00$, and $A = 0.05$. The true outcome distribution is computed using DGP equation 14a and 14b, with the corresponding results displayed in the first row of Table 1 for $A = 0.00$. Subsequently, we list the estimation results (mean, std., and MAE) produced by the Dist-DR, Dist-IPW, and Dist-DML estimators. We list the results for $A = -0.05$ and $A = 0.05$ in Appendix I. We also plot the ground truth and recovered quantile function in Figure 4.

Overall, all estimators are effective in recovering the true outcome distribution. Nonetheless, the Dist-DR estimator yields the largest MAE, and the Dist-IPW estimator offers improved estimations but demonstrates the highest variance. These results are in line with our theoretical analysis. In

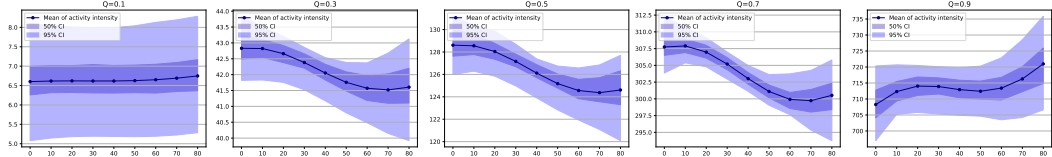

Figure 5: The estimated counterfactual outcome function at quantile 0.1 to 0.9 when working hours ranging from 0 to 80 hours.

contrast, the Dist-DML estimator can correct most of the bias in the Dist-DR estimator and the variance in the Dist-IPW estimator, resulting in more accurate and robust estimates.

## 7    EMPIRICAL APPLICATION

We employ our approach to investigate the causal impact of working hours on physical activity intensity based on a public dataset named the National Health and Nutrition Examination Survey (NHANES), which aims to evaluate the health of people in the United States. The dataset includes demographics, diet, socioeconomics, medical, physiological assessments, and laboratory tests of participants. The physical activity intensity is recorded for successive 1-minute intervals, which constitutes a specific distribution for each person, and we measure it by empirical CDF.

After data preprocessing, we obtain $2,762$ participants. We use the Dist-DML estimator to estimate the causal map, which performs the best in the simulation experiment. We run the experiments 50 times. In each experiment, the estimator is computed 2-fold. Detailed data and statistical descriptions, data preprocessing, and the training details are given in Appendix J.

Figure 5 presents the empirical findings. The lines correspond to the causal map illustrating the distribution of activity intensity at quantiles 0.1, 0.3, 0.5, 0.7, and 0.9 across a range of working hours spanning from 0 to 80 hours per week. The shaded bands represent the 50% and 95% confidence intervals for our estimations.

In general, in the context of regular-level activity intensity (e.g., quantiles lower than 0.7), such as activities like walking and jogging, our analysis reveals a consistent pattern: an increase in working hours is associated with a decrease in activity intensity. This phenomenon can be attributed to the fact that longer working hours tend to displace available time for physical exercise. Conversely, when we focus on high-intensity activities (i.e., activity intensity beyond the 0.9 quantile), our observations suggest an opposite relationship. Specifically, an increase in working hours results in heightened activity intensity. This phenomenon can be attributed to the observation that individuals exhibiting higher levels of activity intensity typically engage in manual labor occupations. Thus, an expansion of working hours among such individuals invariably results in an elevation of their activity intensity levels.

## 8    CONCLUSION

In this paper, we present a novel approach to conducting causal inference in the Wasserstein space, departing from the conventional practice in the Euclidean space. By leveraging Rubin's causal framework, we introduce three estimators: the Dist-DR, Dist-IPW, and Dist-DML estimators, enabling the investigation of the causal impact of continuous treatments on distributional outcomes. Furthermore, we have conducted a comprehensive study of the statistical properties of these estimators, providing valuable theoretical insights. To validate our theoretical findings, we conduct two experiments, one simulation experiment and one empirical application. The results of our study demonstrate the enhanced performance of the Dist-DML estimator. Future research includes i) extending the investigation to other causal estimators, such as ATTE and CATE; ii) exploring the application of this methodology in various domains, including but not limited to healthcare, business, and social sciences.

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

## A    THE DETAILS OF CAUSAL ASSUMPTIONS

As with the majority of previous studies in causal inference Rubin (1978; 2005), our approach relies on four fundamental assumptions. They are *Stable Unit Treatment Unit Assumption* (SUTVA), *Consistency*, *Ignorability*, and *Overlap*.

**Assumption 1 (SUTVA)**  *It contains two parts:*

1. *The potential outcome of a unit is not influenced by the treatment assignment to other units.*

2. *For each unit, there are no different forms of treatment levels that lead to different potential outcomes.*

**Assumption 2 (Consistency)**  *If $A = a$, then $\mathcal{Y} = \mathcal{Y}(a)$.*

**Assumption 3 (Ignorability)**  $A \perp\!\!\!\perp \mathcal{Y}(a) \mid \mathbf{X}$ *for any $a \in \mathcal{A} \subset \mathbb{R}$.*

**Assumption 4 (Overlap)**
*We assume that there exists $c > 0$ such that the conditional probability density function (or generalized propensity score) $p(a|\mathbf{x})$ satisfies*

$$\inf_{a \in \mathcal{A}} \operatorname{ess\,inf}_{\mathbf{x} \in \mathcal{X}} p(a|\mathbf{x}) \geq c.$$

*Furthermore, we assume that $p(a, \mathbf{x})$ is a three-times differentiable function w.r.t. $a$ with all three derivatives being bounded uniformly over the sample space.*

## B    DIFFERENCES FROM THE CLASSICAL FRAMEWORK

The proposed causal framework focuses on the case when the outcome of each sample is a distribution rather than a scalar value. It marks a significant departure from the traditional classical causal framework. This approach opens up new possibilities for causal analysis, especially when dealing with complex datasets and scenarios where scalar representations of outcomes may not adequately capture the underlying heterogeneity in responses.

To further highlight the differences and advantages of our framework, we conduct a comprehensive comparison with the traditional classical causal framework and summarize the results in Table 2 and Figure 6, 7, 8. To distinguish the differences when the realization of the response variable is a scalar or vector, we use $Y$, $Y(a)$, $P_Y(\cdot)$, and $P_{Y(a)}(\cdot)$ to represent the response, the response when $A = a$, the probability measure of $Y$, and the probability measure of $Y(a)$, respectively. Specifically, the main differences can be summarized as three main points.

- **Outcome/Potential outcome variable**. In the classical framework when $A = a$, the realization of potential outcome variable $Y(a)$ is a scalar that is sampled from the potential outcome distribution $P_a(\cdot)$ (Zhou et al., 2022). For example, if $P_a(\cdot) \sim \mathcal{N}(0, 1)$, the corresponding sample is one point drawn from $\mathcal{N}(0, 1)$. In contrast, within our framework, the realization of potential outcome variable $\mathcal{Y}(a)$ is a distribution, which is sampled from a high-dimensional potential outcome distribution $\mathcal{P}_a(\cdot)$. For instance, a realization of $\mathcal{Y}(a)$ could be a normal distribution $\mathcal{N}(\mu, \sigma^2)$, where $(\mu, \log \sigma)$ is a realization drawn from $\mathcal{N}(0, 1)$. Thus, the corresponding sample is a collection of points drawn from $\mathcal{N}(\mu, \sigma^2)$. This comparison is shown in Figure 6.

- **Ambient space of outcome variable** ($\mathbf{\Omega}$). In the classical framework, the realization of the outcome variable is a scalar value with the ambient space as the Euclidean space $\mathbb{R}$. Since the realization of the outcome variable is a distribution in our framework, we consider the ambient space of the outcome variable as $\mathcal{W}_2(\mathcal{I})$ which is the Wasserstein space of distributions over $\mathcal{I}$.

- **Target quantity**. In the classical framework, the essential component is $\mathbb{E}[Y(a)]$ which is a scalar value. Specifically it is the expected value of the response when all units receive treatment $a$. The left hand side (l.h.s) of Figure 7 illustrates the difference of $\mathbb{E}[Y(a)]$ and $\mathbb{E}[Y(\bar{a})]$. In contrast, the response of each unit is characterized as the distribution in our

Table 2: Comparisons between our framework and the framework given in the literature.

| | Our framework | Literature framework |
|---|---|---|
| Treatment/Covariates variable | $A/\mathbf{X}$ | $A/\mathbf{X}$ |
| Outcome/Potential outcome variable | $\mathcal{Y}/\mathcal{Y}(a)$ | $Y/Y(a)$ |
| Ambient space of outcome variable ($\Omega$) | $\mathcal{W}_2(\mathcal{I})$ | $\mathbb{R}$ |
| Probability measure | $\mathcal{P}(\omega), \mathcal{P}_a(\omega)$, where $\omega \in \Omega$ | $P(\omega), P_a(\omega)$, where $\omega \in \Omega$ |
| Metric | Wasserstein | Euclidean |
| Realization of outcome variable | distribution | scalar |
| Target quantity | $\triangle_a, \triangle_{a\bar{a}} \in \mathcal{W}_2(\mathcal{I})$ | $\mathbb{E}[Y(a)], \mathbb{E}[Y(a)] - \mathbb{E}[Y(\bar{a})] \in \mathbb{R}$ |

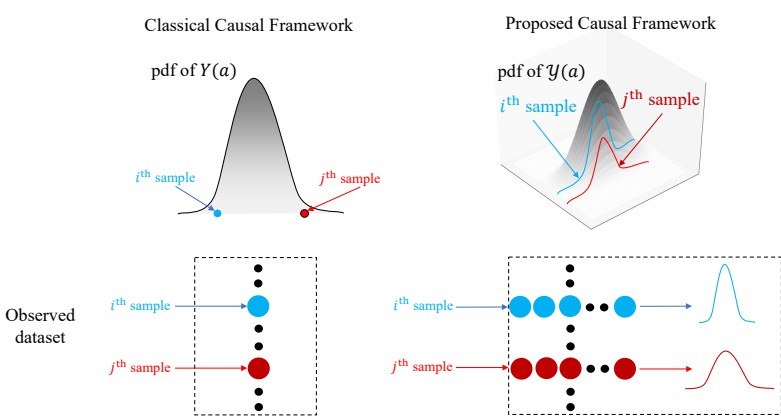

Figure 6: Comparisons between the classical causal framework and our causal framework. In the classical causal framework, the observed dataset contains a finite number of points. Each point represents a realization of a sample. In the proposed casual framework, the observed dataset contains a finite number of collections. Each collection contains finitely many points, and each collection is a realization of a sample.

framework. The essential component is $\triangle_a$ that is the inverse of CDF (also known as the quantile function). Further, $\triangle_a$ represents the quantile function of the barycenter in $\mathcal{W}_2(\mathcal{I})$ provided all units receive $A = a$. The subtraction between $\triangle_a$ and $\triangle_{\bar{a}}$ (denoted as $\triangle_{a\bar{a}}$) is also called the *quantile differences of causal effect map* as it represents the difference at various quantiles. Figure 8 displays the quantities $\triangle_a$, $\triangle_{\bar{a}}$, and $\triangle_{a\bar{a}}$ visually. By characterizing causal effects as distributions, we gain a more comprehensive understanding of the entire distribution of potential outcomes.

## C  PROOF OF PROPOSITION 1

**Proof 1** *If we can prove that $\mathbb{E}[\mathcal{Y}(a)^{-1}] = \mu_a^{-1}$, then we have $\triangle_a = \mu_a^{-1} = \mathbb{E}[\mathcal{Y}(a)^{-1}]$. Let $\mathcal{Q}$ be the set containing all the left-continuous non-decreasing functions on $(0,1)$. If we view $\mathcal{Q}$ as a subspace of $L^2([0,1])$, then it is isometric to $\mathcal{W}_2(\mathcal{I})$ (e.g., see Panaretos & Zemel (2020)). Indeed,*

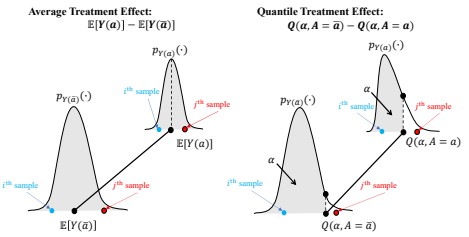

Figure 7: ATE and QTE in the literature.

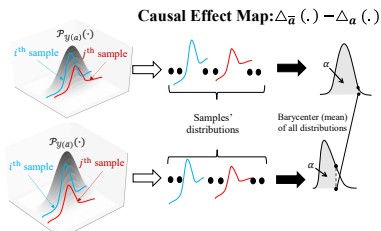

Figure 8: Causal Effect Map in our paper.

$\mu_a = \underset{\nu \in \mathcal{W}_2(\mathcal{I})}{\arg\min} \mathbb{E}\big[\mathbb{D}_2(\mathcal{Y}(a), \nu)^2\big] \overset{\bullet}{=} \underset{\nu \in \mathcal{Q}}{\arg\min} \ \mathbb{E}\big[\int_0^1 |\mathcal{Y}(a)^{-1}(t) - \nu^{-1}(t)|^2 dt\big]$. *Here,* $\overset{\bullet}{=}$ *follows from Theorem 2.18 of Villani (2021). Since we can interchange the integral sign $\int$ and $\mathbb{E}$, we notice that* $\mathbb{E}\big[\int_0^1 |\mathcal{Y}(a)^{-1}(t) - \nu^{-1}(t)|^2 dt\big] = \int_0^1 \mathbb{E}\big[|\mathcal{Y}(a)^{-1}(t) - \nu^{-1}(t)|^2\big] dt = \int_0^1 (\mathbb{E}\big[\mathcal{Y}(a)^{-1}(t)\big] - \nu^{-1}(t))^2 dt + \int_0^1 \mathbb{E}[(\mathbb{E}\big[\mathcal{Y}(a)^{-1}(t)\big] - \mathcal{Y}(a)^{-1}(t))^2] dt,$ *and* $\mathbb{E}\big[\int_0^1 |\mathcal{Y}(a)^{-1}(t) - \nu^{-1}(t)|^2 dt\big]$ *attains its minimum when* $\nu^{-1}(t) = \mathbb{E}\big[\mathcal{Y}(a)^{-1}(t)\big]$. *We can therefore conclude that* $\mu_a = \big(\mathbb{E}\big[\mathcal{Y}(a)^{-1}\big]\big)^{-1}$.

## D    PROOF OF PROPOSITION 2

Heuristically, the Delta Dirac function $\delta(x)$ is defined such that $\delta(x) = \begin{cases} \infty & x = 0 \\ 0 & x \neq 0 \end{cases}$ and $\int_{\mathbb{R}} \delta(x) dx = 1$. Furthermore, for any continuous function $f$ defined on $\Omega$ such that $0 \in \Omega$, we have $\int_{\Omega} f(x)\delta(x) dx = f(0)$. Finally, we denote $\delta_0(\cdot) = \delta(\cdot)$ and $\delta_a(x) = \delta(x - a)$.

**Proof 2** *Indeed, we have*

$$\mathbb{E}\left[\frac{\delta_a(A)}{p(a|\mathbf{X})}\mathcal{Y}^{-1}\right] = \mathbb{E}\left[\frac{1}{p(a|\mathbf{X})}\mathbb{E}[\delta_a(A)\mathcal{Y}^{-1}|\mathbf{X}]\right]$$

$$=\mathbb{E}\left[\frac{1}{p(a|\mathbf{X})}\int_{\bar{a}\in\mathcal{A}}\mathbb{E}[\delta_a(A)\mathcal{Y}^{-1}|A=\bar{a},\mathbf{X}]p(\bar{a}|\mathbf{X})d\bar{a}\right]$$

$$=\mathbb{E}\left[\frac{1}{p(a|\mathbf{X})}\int_{\bar{a}\in\mathcal{A}}\delta_a(\bar{a})\mathbb{E}[\mathcal{Y}^{-1}|A=\bar{a},\mathbf{X}]p(\bar{a}|\mathbf{X})d\bar{a}\right] = \mathbb{E}\left[\frac{1}{p(a|\mathbf{X})}\mathbb{E}[\mathcal{Y}^{-1}|A=a,\mathbf{X}]p(a|\mathbf{X})\right]$$

$$=\mathbb{E}[\mathbb{E}[\mathcal{Y}^{-1}|A=a,\mathbf{X}]] \overset{\star}{=} \mathbb{E}[\mathbb{E}[\mathcal{Y}(a)^{-1}|A=a,\mathbf{X}]] \overset{*}{=} \mathbb{E}[\mathbb{E}[\mathcal{Y}(a)^{-1}|\mathbf{X}]] = \mathbb{E}[\mathcal{Y}(a)^{-1}] = \triangle_a.$$

*Again,* $\star$ *is due to Causal Assumption (2) and* $*$ *is due to Causal Assumption (3).*

*In Table 3, we present some commonly used second-order kernels found in the literature.*

Table 3: Some common kernel functions of order 2 that exist in the literature

|  | Kernel Function $K(u)$ | Support |
|---|---|---|
| Uniform | $K(u) = \frac{1}{2}$ | $|u| \leq 1$ |
| Triangular | $K(u) = (1 - |u|)$ | $|u| \leq 1$ |
| Epanechnikov | $K(u) = \frac{3}{4}(1 - u^2)$ | $|u| \leq 1$ |
| Quartic | $K(u) = \frac{15}{16}(1 - u^2)^2$ | $|u| \leq 1$ |
| Triweight | $K(u) = \frac{35}{32}(1 - u^2)^3$ | $|u| \leq 1$ |
| Tricube | $K(u) = \frac{70}{81}(1 - |u|^3)^3$ | $|u| \leq 1$ |
| Gaussian | $K(u) = \frac{1}{\sqrt{2\pi}}e^{-\frac{u^2}{2}}$ | $u \in \mathbb{R}$ |
| Cosine | $K(u) = \frac{\pi}{4}\cos\left(\frac{\pi}{2}u\right)$ | $|u| \leq 1$ |
| Logistic | $K(u) = \frac{1}{e^u + 2 + e^{-u}}$ | $u \in \mathbb{R}$ |
| Sigmoid | $K(u) = \frac{2}{\pi}\frac{1}{e^u + e^{-u}}$ | $u \in \mathbb{R}$ |

## E    PROOF OF PROPOSITION 3

**Proof 3** *We have proven that* $\mathbb{E}[m_a(\mathbf{X})] = \triangle_a$ *in equation 3 under given Assumptions. Additionally, we have proven that* $\mathbb{E}\left[\frac{\delta_a(A)}{p(a|\mathbf{X})}\mathcal{Y}^{-1}\right]$ *in Proposition 2. It suffices to prove that*

$\mathbb{E}\left[\frac{\delta_a(A)}{p(a|\mathbf{X})}m_a(\mathbf{X})\right] = \triangle_a$. *Indeed, we have*

$$\mathbb{E}\left[\frac{\delta_a(A)}{p(a|\mathbf{X})}m_a(\mathbf{X})\right] = \mathbb{E}\left[\frac{m_a(\mathbf{X})}{p(a|\mathbf{X})}\mathbb{E}[\delta_a(A)|\mathbf{X}]\right]$$

$$=\mathbb{E}\left[\frac{m_a(\mathbf{X})}{p(a|\mathbf{X})}\int_{\bar{a}\in\mathcal{A}}\delta_a(\bar{a})p(\bar{a}|\mathbf{X})d\bar{a}\right] = \mathbb{E}\left[\frac{m_a(\mathbf{X})}{p(a|\mathbf{X})}p(a|\mathbf{X})\right] = \mathbb{E}[m_a(\mathbf{X})] = \triangle_a.$$

*The proof is completed.*

## F  PROOF OF DOUBLY ROBUST PROPERTY

**Proof 4** *We need to show that, if either $m_a(\mathbf{X})$ or $\mathbb{P}\{A = a|\mathbf{X}\}$ is misspecified as $\bar{m}_a(\mathbf{X})$ or $\bar{\mathbb{P}}\{A = a|\mathbf{X}\}$ accordingly, then*

$$\mathbb{E}\left[\bar{m}_a(\mathbf{X}) + \frac{\delta(A - a)}{\mathbb{P}\{A = a|\mathbf{X}\}}(\mathcal{Y}^{-1} - \bar{m}_a(\mathbf{X}))\right] = \triangle_a \quad and \tag{15a}$$

$$\mathbb{E}\left[m_a(\mathbf{X}) + \frac{\delta(A - a)}{\bar{\mathbb{P}}\{A = a|\mathbf{X}\}}(\mathcal{Y}^{-1} - m_a(\mathbf{X}))\right] = \triangle_a. \tag{15b}$$

*Indeed, equation 15a follows from equation 5 and the following derivations:*

$$\mathbb{E}\left[\frac{\delta(A - a)}{\mathbb{P}\{A = a|\mathbf{X}\}}\bar{m}_a(\mathbf{X})\right] = \mathbb{E}\left[\frac{\bar{m}_a(\mathbf{X})}{\mathbb{P}\{A = a|\mathbf{X}\}}\mathbb{E}[\delta(A - a)|\mathbf{X}]\right] = \mathbb{E}[\bar{m}_a(\mathbf{X})].$$

*On the other hand, equation 15a follows from equation 3 and the following derivations:*

$$\mathbb{E}\left[\frac{\delta(A - a)}{\bar{\mathbb{P}}\{A = a|\mathbf{X}\}}[\mathcal{Y}^{-1} - m_a(\mathbf{X})]\right] = \sum_{\bar{a}\in\mathcal{A}}\mathbb{E}\left[\frac{\mathbb{E}[\delta(A - a)[\mathcal{Y}^{-1} - m_a(\mathbf{X})]|A = \bar{a}, \mathbf{X}]\mathbb{P}\{A = \bar{a}|\mathbf{X}\}}{\bar{\mathbb{P}}\{A = a|\mathbf{X}\}}\right]$$

$$=\mathbb{E}\left[\frac{(\mathbb{E}[\mathcal{Y}^{-1}|A = a, \mathbf{X}] - m_a(\mathbf{X}))\mathbb{P}\{A = a|\mathbf{X}\}}{\bar{\mathbb{P}}\{A = a|\mathbf{X}\}}\right] = 0.$$

## G  ALGORITHM FOR CROSS-FITTED ESTIMATORS

---

**Algorithm 1** Computations of $\hat{\triangle}_{a;w}$, where $w \in \{DR, IPW, DML\}$

---

**Require:** Realizations of $(A_s, \mathbf{X}_s, \mathcal{Y}_s)_{s=1}^N$. Determine the kernel function $K(\cdot)$.
1: Estimate $\hat{\mathcal{Y}}^{-1}$ for each sample.
2: Split $(A_s, \mathbf{X}_s, \hat{\mathcal{Y}}_s)_{s=1}^N$ to $\mathcal{K}$ disjoint samples $\mathcal{D}_k$ where $k \in \{1, \cdots, \mathcal{K}\}$ and formulate $\mathcal{D}_{-k}$. The size of $\mathcal{D}_k$ is $N_k$.
3: **for** $k \leftarrow 1$ to $\mathcal{K}$ **do**
4:    Estimate $\hat{p}^k(a|\cdot)$ based on $\mathcal{D}_{-k}$.
5:    Estimate $\hat{m}_a(\cdot)$ based on $\mathcal{D}_{-k}$.
6:    Compute $\hat{\triangle}_{a;DR}^k$, $\hat{\triangle}_{a;IPW}^k$, and $\hat{\triangle}_{a;DML}^k$ according to equation 9, 10, and 11.
7: **end for**
8: Compute $\hat{\triangle}_{a;w}$ according to equation 12.

---

## H  PROOF OF THEOREM 1

Theorem 1 only gives the asymptotic property of $\hat{\triangle}_{a;DML}$. Indeed, we can also study the asymptotic properties of $\hat{\triangle}_{a;DR}$ and $\hat{\triangle}_{a;IPW}$. Before presenting the proofs of Theorem 1, let's give the asymptotic properties of $\hat{\triangle}_{a;w}$ where $w \in \{DR, IPW, DML\}$.

**Theorem 2** *Suppose that $p(a|\mathbf{x}) \in \mathcal{C}^3$ on $\mathcal{A}$ such that the derivatives (including $0$ order derivative) is bounded uniformly in the sample space for any $\mathbf{x}$. Further, we assume that $\mathbb{E}[\mathcal{Y}^{-1}|A = a, \mathbf{X}] \in \mathcal{C}^3$ on $[0,1] \times \mathcal{A}$ and $\mathbb{E}[\|\mathcal{Y}^{-1}\| | A = a, \mathbf{X}] \in \mathcal{C}^3$ on $\mathcal{A}$ which are bounded uniformly in the sample spaces. Then we have*

$$\sqrt{N}(\hat{\triangle}_{a;DR} - \triangle_a) = \sqrt{N}[(\mathbb{P}_N - \mathbb{E})m_a(\mathbf{X})] + o_P(1) \tag{16}$$

*when $\rho_m(N^{-\frac{1}{2}})$. Further, for any $w \in \{IPW, DML\}$, if $h \to 0$, $Nh \to \infty$, and $Nh^5 \to C \in [0, \infty)$, then, under the convergence assumptions, we have*

$$\sqrt{Nh}(\hat{\triangle}_{a;w} - \triangle_a) = \sqrt{Nh}\left[\mathbb{P}_N\{\varphi(A, \mathbf{X}, \mathcal{Y})\} - \triangle_a\right] + o_P(1), \tag{17}$$

1. *where $\varphi(A, \mathbf{X}, \mathcal{Y}) = \frac{K_h(A=a)\mathcal{Y}^{-1}}{p(a|\mathbf{X})}$ if $w = IPW$ and $\rho_p = o(N^{-\frac{1}{2}})$;*

2. *where $\varphi(A, \mathbf{X}, \mathcal{Y}) = \frac{K_h(A-a)\{\mathcal{Y}^{-1} - m_a(\mathbf{X})\}}{p(a|\mathbf{X})} + m_a(\mathbf{X})$ if $w = DML$ and $\rho_m \rho_p = o(N^{-\frac{1}{2}})$, $\rho_m = o(1)$, $\rho_p = o(1)$.*

*Additionally,*

$$\sqrt{Nh}\{\hat{\triangle}_{a;w} - \triangle_a - h^2 B_a\} \tag{18a}$$

*converges weakly to a centred Gaussian process in $\mathcal{L}^2([0,1])$ where*

$$B_a = \left(\int u^2 K(u)du\right) \times$$

$$\begin{cases} \frac{1}{2}\mathbb{E}\left[\frac{m_a(\mathbf{X})\partial_{aa}^2 p(a|\mathbf{X})}{p(a|\mathbf{X})}\right] + \mathbb{E}\left[\frac{\partial_a m_a(\mathbf{X})\partial_a p(a|\mathbf{X})}{p(a|\mathbf{X})}\right] + \frac{1}{2}\mathbb{E}[\partial_{aa}^2 m_a(\mathbf{X})] & \text{if } w = IPW \\ \mathbb{E}\left[\frac{\partial_a m_a(\mathbf{X})\partial_a p(a|\mathbf{X})}{p(a|\mathbf{X})}\right] + \frac{1}{2}\mathbb{E}[\partial_{aa}^2 m_a(\mathbf{X})] & \text{if } w = DML. \end{cases} \tag{18b}$$

In lieu of proving Theorem 1, we prove Theorem 2. Nevertheless, the proof of Theorem 2 requires two Lemmas.

**Lemma 1** *For $G_1$, $G_2 \in \mathcal{W}_2(\mathcal{I})$, we have $\|G_1 - G_2\| = \mathbb{D}_2(G_1, G_2)$.*

**Lemma 2** *Under Convergence Assumption 1, we have $\frac{1}{N}\sum_{s=1}^{N}\|\hat{\mathcal{Y}}_s^{-1} - \mathcal{Y}_s^{-1}\|^2 = O_P(\alpha_N^2 + \nu_N^2)$.*

**Proof 5 (Proof of Theorem 1)** *In the following proof, we first show that the case when the estimator is $\hat{\triangle}_{a;DR}$. Generally, we assume that $K = 2$. The general case is similar. For simplicity, we define four operators $\mathbb{P}_N$, $\mathbb{P}_{N_k}$, $\mathbb{E}_N$, and $\mathbb{E}_{N_k}$ such that given a random quantity $\mathcal{O}$, $\mathbb{P}_N\mathcal{O} = \frac{1}{N}\sum_{s=1}^{N}\mathcal{O}_s$, $\mathbb{P}_{N_k} = \frac{1}{N_k}\sum_{s \in \mathcal{D}_k}\mathcal{O}_s$, $\mathbb{E}_N = \frac{1}{N}\sum_{s=1}^{N}\mathbb{E}[\mathcal{O}_s]$, and $\mathbb{E}_{N_k} = \frac{1}{N_k}\sum_{s \in \mathcal{D}_k}\mathbb{E}[\mathcal{O}_s]$. Given the distributions $\lambda$. Define $\mathcal{L}\lambda = \lambda^{-1}$. Let $Z_s = \mathcal{L}\mathcal{Y}_s$, and if the $s^{\text{th}}$ subject belongs to the $k$ partition, then $\hat{Z}_s = \mathcal{L}\hat{\mathcal{Y}}_s$ and $R_s = \hat{Z}_s - Z_s$. Define $D_a^k(\cdot) = \hat{m}_a^k(\cdot) - \tilde{m}_a^k(\cdot)$. Under the causal assumptions, we can show that $\triangle_a = \psi_a = \mathbb{E}[m_a(\mathbf{X})]$. Denote the corresponding sampled version using $\mathcal{D}_k$ as $\hat{\triangle}_{a;DR}^k = \mathbb{P}_{N_k}[\hat{m}_a^k(\mathbf{X})]$. As a result, we have the cross-fitting estimator $\hat{\triangle}_{a;DR}$ such that*

$$\hat{\triangle}_{a;DR} = \sum_{k=1}^{2}\frac{N_k}{N}\hat{\triangle}_{a;DR}^k = \frac{1}{N}(N_1\hat{\triangle}_{a;DR}^{D;1} + N_2\hat{\triangle}_{a;DR}^2).$$

*Next, we consider the difference $\sqrt{N}(\hat{\triangle}_{a;DR} - \triangle_a)$. Indeed, we have*

$$\sqrt{N}\left[\frac{1}{N}(N_1\hat{\triangle}_{a;DR}^1 + N_2\hat{\triangle}_{a;DR}^2) - \triangle_a\right] = \sqrt{N}\left[\frac{1}{N}\sum_{k=1,2}N_k\mathcal{A}_k - \psi_a\right],$$

where $\mathcal{A}_k = \mathbb{P}_{N_k}[(\tilde{m}_a^k(\mathbf{X}) + D_a^k(\mathbf{X}))]$. We can then decompose $\sqrt{N}\left[\frac{1}{N}\sum_{k=1,2} N_k \mathcal{A}_k - \psi_a\right]$ into the sum of five quantities as follows:

$$\frac{\sqrt{N}}{N}\sum_{k=1,2} N_k(I + II + III + IV + V)$$

where

$$
\begin{aligned}
I &= (\mathbb{P}_{N_k} - \mathbb{E}_{N_k})[\tilde{m}_a^k(\mathbf{X}) - m_a(\mathbf{X})] \\
II &= (\mathbb{P}_{N_k} - \mathbb{E}_{N_k})[m_a(\mathbf{X})] = (\mathbb{P}_{N_k} - \mathbb{E}_{N_k})\varphi(A, \mathbf{X}, \mathcal{Y}) \\
III &= \mathbb{E}_{N_k}[(\tilde{m}_a^k(\mathbf{X}) - m_a(\mathbf{X}))] \\
IV &= \mathbb{P}_{N_k}\{D_a^k(\mathbf{X})\}.
\end{aligned}
$$

The proof follows from the Slutsky's Lemma after we get the bounds of I, III, and IV.

Boundedness of I: Let

$$H(A, \mathbf{X}, Z) = \tilde{m}_a^k(\mathbf{X}) - m_a(\mathbf{X}).$$

Hence, we have $\mathbb{E}[\|I\|^2] = \mathbb{E}[\|(\mathbb{P}_{N_k} - \mathbb{E}_{N_k})H(A, \mathbf{X}, Z)\|^2]$. We now simplify the quantity $\mathbb{E}[\|(\mathbb{P}_{N_k} - \mathbb{E}_{N_k})H(A, \mathbf{X}, Z)\|^2]$. Indeed, we have

$$\mathbb{E}[\|(\mathbb{P}_{N_k} - \mathbb{E}_{N_k})H(A, \mathbf{X}, Z)\|^2] = \frac{1}{N_k^2}\mathbb{E}[\|\sum_{s \in \mathcal{D}_k}\{H(A_s, \mathbf{X}_s, Z_s) - \mathbb{E}[H(A_s, \mathbf{X}_s, Z_s)]\}\|^2]$$

$$= \frac{1}{N_k^2}\sum_{s \in \mathcal{D}_k}\mathbb{E}[\|H(A_s, \mathbf{X}_s, Z_s) - \mathbb{E}[H(A_s, \mathbf{X}_s, Z_s)]\|^2] + \frac{1}{N_k^2}\sum_{\substack{s,\bar{s} \in \mathcal{D}_k \\ s \neq \bar{s}}} C_{s\bar{s}} := I_1 + I_2,$$

where $C_{s\bar{s}} = \mathbb{E}[\langle H_s - \mathbb{E}[H_s], H_{\bar{s}} - \mathbb{E}[H_{\bar{s}}]\rangle]$ and $H_s = H(A_s, \mathbf{X}_s, Z_s)$. Consider the term $I_1$. We have

$$I_1 \lesssim \frac{1}{N_k^2}\sum_{s \in \mathcal{D}_k}\mathbb{E}[\|H(A_s, \mathbf{X}_s, Z_s)\|^2].$$

We can bound $\mathbb{E}[\|H(A_s, \mathbf{X}_s, Z_s)\|^2]$. Indeed, we have

$$\mathbb{E}[\|H(A_s, \mathbf{X}_s, Z_s)\|^2] = \mathbb{E}[\|\tilde{m}_a^k(\mathbf{X}) - m_a(\mathbf{X})\|^2] \leq \rho_m^2.$$

As a result, we have

$$I_1 \lesssim \frac{1}{N_k}\rho_m^2 \lesssim \frac{1}{N}\rho_m^2.$$

We now consider the quantity $I_2$. Note that

$$\mathbb{E}[\langle H_s - \mathbb{E}[H_s], H_{\bar{s}} - \mathbb{E}[H_{\bar{s}}]\rangle] = \mathbb{E}[\langle H_s, H_{\bar{s}}\rangle] - \langle\mathbb{E}[H_s], \mathbb{E}[H_{\bar{s}}]\rangle \lesssim \|\mathbb{E}[H_s]\| \times \|\mathbb{E}[H_{\bar{s}}]\|.$$

Since

$$\|\mathbb{E}[H_s]\| \lesssim \mathbb{E}[\|m_a(\mathbf{X}) - \tilde{m}_a^k(\mathbf{X})\|] \leq \rho_m.$$

Hence, we have $C_{s\bar{s}} \lesssim \rho_m^2$ and $I_2 \lesssim \left(1 - \frac{1}{N_k}\right)\rho_m^2$. As a result, we can show that

$$\mathbb{E}[\|I\|^2] = O(N^{-1}\rho_m^2 + \rho_m^2).$$

Thus, we have $I = O_P(N^{-\frac{1}{2}}\rho_m + \rho_m)$.

Boundedness of III: For simplicity, we denote

$$\mathcal{A} = \mathbb{E}_{N_k}[(\tilde{m}_a^k(\mathbf{X}) - m_a(\mathbf{X}))].$$

We consider the quantity $\mathbb{E}[\|\mathcal{A}\|]$. Since $\mathcal{A}$ is an expectation already, we have $\mathbb{E}[\|\mathcal{A}\|] = \|\mathcal{A}\|$. Further, we can simplify $\|\mathcal{A}\|$ as follows:

$$\|\mathcal{A}\| = \|\mathbb{E}[(\tilde{m}_a^k(\mathbf{X}) - m_a(\mathbf{X}))]\| \leq \mathbb{E}[\|(\tilde{m}_a^k(\mathbf{X}) - m_a(\mathbf{X}))\|] \leq \rho_m.$$

Boundedness of IV: Let $\mathcal{A} = \mathbb{P}_{N_k}\{D_a^k(\mathbf{X})\}$. Consider $\|\mathcal{A}\|^2$. We have

$$\|\mathcal{A}\|^2 = \|\mathbb{P}_{N_k}\{D_a^k(\mathbf{X})\}\|^2 = \underbrace{\frac{1}{N_k^2}\sum_{s \in \mathcal{D}_k}\|D_a^k(\mathbf{X}_s)\|^2}_{IV_1} + \underbrace{\frac{1}{N_k^2}\sum_{\substack{s,\bar{s} \in \mathcal{D}_k \\ s \neq \bar{s}}}\langle D_{i,k}(\mathbf{X}_s), D_a^k(\mathbf{X}_{\bar{s}})\rangle}_{IV_2}.$$

*Consider $IV_1$ first. Using Assumption 1, we see that $IV_1 \leq \frac{C}{N_k^2} \sum_{s \in \mathcal{D}_k} \|D_a^k(\mathbf{X}_s)\|^2$ for some constant $C$. Note that, for any $\delta > 0$, we have*

$$\mathbb{P}\Big\{ \frac{1}{N_k} \sum_{s \in \mathcal{D}_k} \|D_a^k(\mathbf{X}_s)\|^2 \geq \frac{\|\hat{m}_a^k - \tilde{m}_a^k\|^2}{\delta} \Big\} \leq \frac{\delta \mathbb{E}\Big[ \frac{1}{N_k} \sum_{s \in \mathcal{D}_k} \|D_a^k(\mathbf{X}_s)\|^2 \Big]}{\|\hat{m}_a^k - \tilde{m}_a^k\|^2} = \frac{\delta \mathbb{E}[\|D_a^k(\mathbf{X})\|^2]}{\|\hat{m}_a^k - \tilde{m}_a^k\|^2} = \delta.$$

*Indeed, the inequality follows from Markov inequality. The last equality follows from the Definition of $\|\cdot\|^2$. According to the definition, we have $\mathbb{E}[\|D_a^k(\mathbf{X})\|^2] = \|\hat{m}_a^k - \tilde{m}_a^k\|^2$. It means that $\frac{1}{N_k} \sum_{s \in \mathcal{D}_k} \|D_a^k(\mathbf{X}_s)\|^2 = O_P(\|\hat{m}_a^k - \tilde{m}_a^k\|^2)$. Hence, we note that $IV_1 = \frac{1}{N_k} \times \frac{1}{N_k} \sum_{s \in \mathcal{D}_k} \|D_a^k(\mathbf{X}_s)\|^2 = \frac{N}{N_k} \times \frac{1}{N} \times O_P(\|\hat{m}_a^k - \tilde{m}_a^k\|^2)$. Using Convergence Assumptions 2 and 3, we have $IV_1 = O_P(N^{-2} + N^{-1}\alpha_N^2 + N^{-1}\nu_N^2)$. Next, we consider $IV_2$. Let*

$$\mathcal{A} = (\mathbb{E}[\|D_a^k(\mathbf{X}_s)\|^4])^{\frac{1}{4}} (\mathbb{E}[\|D_a^k(\mathbf{X}_{\bar{s}})\|^4])^{\frac{1}{4}}.$$

*Note that, for any $\delta > 0$, we have*

$$\mathbb{P}\Big\{ IV_2 \geq \frac{\mathcal{A}}{\delta} \Big\} \leq \frac{\delta \frac{1}{N_k^2} \sum_{\substack{s,\bar{s} \in \mathcal{D}_k \\ s \neq \bar{s}}} \mathbb{E}\Big[ \langle D_a^k(\mathbf{X}_s), D_a^k(\mathbf{X}_{\bar{s}}) \rangle \Big]}{\mathcal{A}} \overset{\star}{\leq} \frac{\delta \frac{1}{N_k^2} \sum_{\substack{s,\bar{s} \in \mathcal{D}_k \\ s \neq \bar{s}}} \mathcal{A}}{\mathcal{A}} = \frac{\delta N_k(N_k - 1)}{N_k^2} = \delta\Big(1 - \frac{1}{N_k}\Big) \leq \delta.$$

*Here, $\overset{\star}{\leq}$ is due to the upper bound of the quantity $\mathbb{E}\big[ \langle D_a^k(\mathbf{X}_s), D_a^k(\mathbf{X}_{\bar{s}}) \rangle \big]$. Indeed, using the fact that the unit $s$ and the unit $\bar{s}$ are independent of each other, we have*

$$\mathbb{E}[\langle D_a^k(\mathbf{X}_s), D_a^k(\mathbf{X}_{\bar{s}}) \rangle] \lesssim (\mathbb{E}[\langle D_a^k(\mathbf{X}_s), D_a^k(\mathbf{X}_{\bar{s}}) \rangle^2])^{\frac{1}{2}} (\mathbb{E}[\|D_a^k(\mathbf{X}_s)\|^2 \|D_a^k(\mathbf{X}_{\bar{s}})\|^2])^{\frac{1}{2}}$$

$$\leq (\mathbb{E}[\|D_a^k(\mathbf{X}_s)\|^4])^{\frac{1}{4}} (\mathbb{E}[\|D_a^k(\mathbf{X}_{\bar{s}})\|^4])^{\frac{1}{4}} = (\mathbb{E}[\|D_a^k(\mathbf{X})\|^4])^{\frac{1}{2}} = O_P(\|\hat{m}_a^k - \tilde{m}_a^k\|^2).$$

*Hence, we can conclude that $IV_2 = O_P(N^{-1} + \alpha_N^2 + \nu^2)$. Consequently, we obtain that $\|IV\|^2 = O_P(N^{-2} + N^{-1}\alpha_N^2 + N^{-1}\nu_N^2)$, implying that $IV = O_P(N^{-1} + N^{-\frac{1}{2}}\alpha_N + N^{-\frac{1}{2}}\nu_N)$. Finally, $\frac{\sqrt{N}}{N} \sum_{k=1,2} N_k II$ converges weakly to a centred Gaussian process due to the Central Limit Theorem. The proof is completed.*

*Next, we prove the case when the estimators are chosen as $\hat{\triangle}_{a;w}$, where $w \in \{IPW, DML\}$. We only present the proofs for the estimator $\hat{\triangle}_{a;DML}$. To prove the results for the estimator $\hat{\triangle}_{a;IPW}$, we only need to replace the terms $m_a(\mathbf{X})$, $\hat{m}_a^k(\mathbf{X})$, and $D_a^k$ in the following proof with 0.*

*Again, we consider the case when $K = 2$ for simplicity; the general case can be proven in a similar fashion. Let $Z = \mathcal{L}\mathcal{Y}$ and $\hat{Z} = \mathcal{L}\hat{\mathcal{Y}}$, where $\mathcal{L}\mathcal{Y} = \mathcal{Y}^{-1}$. Write $R_i = \hat{Z}_i - Z_i$ and $D_a^k(\mathbf{x}) = \hat{m}_a^k(\mathbf{x}) - \tilde{m}_a^k(\mathbf{x})$. Define*

$$\psi_a = \mathbb{E}\Big[ \frac{K_h(A - a)Z}{p(a|\mathbf{X})} - \big\{ \frac{K_h(A - a)}{p(a|\mathbf{X})} - 1 \big\} m_a(\mathbf{X}) \Big], \tag{19}$$

$$\hat{\psi}_{a,k} = \mathbb{P}_{N_k}\Big[ \frac{K_h(A - a)\hat{Z}}{\hat{p}^k(a|\mathbf{X})} - \big\{ \frac{K_h(A - a)}{\hat{p}^k(a|\mathbf{X})} - 1 \big\} \hat{m}_a^k(\mathbf{X}) \Big]. \tag{20}$$

*Hence, we have*

$$\hat{\triangle}_{a;DML} = \frac{1}{N}(N_1 \hat{\psi}_{a,1} + N_2 \hat{\psi}_{a,2}).$$

*Moreover, since $h \to 0$, W.L.O.G., we assume that $h < 1$. Hence, we have $0 < \sqrt{h} < 1$ and $0 < \frac{\sqrt{Nh}}{N} < \frac{\sqrt{N}}{N} = \frac{1}{\sqrt{N}}$. Note that from Eqn. equation 19, we have*

$$\psi_a = \mathbb{E}\Big[ \frac{K_h(A - a)(Z - m_a(\mathbf{X}))}{p(a|\mathbf{X})} \Big] + \triangle_a.$$

*As a result, we have*

$$\sqrt{Nh}\big( \hat{\triangle}_{a;DML} - \triangle_a \big) = \sqrt{Nh}\big( \frac{1}{N}(N_1 \hat{\psi}_{a,1} + N_2 \hat{\psi}_{a,2}) - \triangle_a \big)$$

$$= \sqrt{Nh}\big( \frac{1}{N}(N_1 \hat{\psi}_{a,1} + N_2 \hat{\psi}_{a,2}) - \psi_a \big) + \sqrt{Nh}\mathbb{E}\Big[ \frac{K_h(A - a)(Z - m_a(\mathbf{X}))}{p(a|\mathbf{X})} \Big].$$

*We can then decompose $\sqrt{Nh}\left(\frac{1}{N}(N_1\hat{\psi}_{a,1} + N_2\hat{\psi}_{a,2}) - \psi_a\right)$ into the sum of five terms as follows:*

$$\sqrt{Nh}\left(\frac{1}{N}(N_1\hat{\psi}_{a,1} + N_2\hat{\psi}_{a,2}) - \psi_a\right)$$
$$=\sqrt{N}\sum_{k=1,2}\frac{N_k}{N}I + \sqrt{N}\sum_{k=1,2}\frac{N_k}{N}II + \sqrt{N}\sum_{k=1,2}\frac{N_k}{N}III + \sqrt{N}\sum_{k=1,2}\frac{N_k}{N}IV + \sqrt{N}\sum_{k=1,2}\frac{N_k}{N}V,$$

*where*

$$I = \sqrt{h}(\mathbb{P}_{N_k} - \mathbb{E}_{N_k})\left[\frac{K_h(A-a)(Z - \tilde{m}_a^k(\mathbf{X}))}{\hat{p}^k(a|\mathbf{X})} + \tilde{m}_a^k(\mathbf{X}) - \frac{K_h(A-a)(Z - m_a(\mathbf{X}))}{p(a|\mathbf{X})} - m_a(\mathbf{X})\right]$$

$$II = \sqrt{h}(\mathbb{P}_{N_k} - \mathbb{E}_{N_k})\left[\frac{K_h(A-a)(Z - m_a(\mathbf{X}))}{p(a|\mathbf{X})} + m_a(\mathbf{X})\}\right] = \sqrt{h}(\mathbb{P}_{N_k} - \mathbb{E}_{N_k})\varphi(A, \mathbf{X}, \mathcal{Y})$$

$$III = \sqrt{h}\mathbb{E}_{N_k}\left[K_h(A-a)(Z - m_a(\mathbf{X}))\frac{(p(a|\mathbf{X}) - \hat{p}^k(a|\mathbf{X}))}{\hat{p}^k(a|\mathbf{X})p(a|\mathbf{X})}\right]$$
$$+ \sqrt{h}\mathbb{E}_{N_k}\left[\frac{\{\tilde{m}_a^k(\mathbf{X}) - m_a(\mathbf{X})\}\{\hat{p}^k(a|\mathbf{X}) - K_h(A-a)\}}{\hat{p}^k(a|\mathbf{X})}\right]$$

$$IV = \sqrt{h}\mathbb{P}_{N_k}\left[\left\{1 - \frac{K_h(A-a)}{\hat{p}^k(a|\mathbf{X})}\right\}\{D_a^k(\mathbf{X})\}\right], \quad V = \sqrt{h}\mathbb{P}_{N_k}\left[\frac{K_h(A-a)R}{\hat{p}^k(a|\mathbf{X})}\right].$$

*Define*

$$H_1(A, \mathbf{X}, Z) = \frac{K_h(A-a)Z\{p(a|\mathbf{X}) - \hat{p}^k(a|\mathbf{X})\}}{\hat{p}^k(a|\mathbf{X})p(a|\mathbf{X})}$$
$$H_2(A, \mathbf{X}, Z) = \frac{K_h(A-a)\{\hat{p}^k(a|\mathbf{X})m_a(\mathbf{X}) - p(a|\mathbf{X})\tilde{m}_a^k(\mathbf{X})\}}{\hat{p}^k(a|\mathbf{X})p(a|\mathbf{X})}$$
$$H_3(A, \mathbf{X}, Z) = \tilde{m}_{a,k}(\mathbf{X}) - m_a(\mathbf{X})$$
$$H(A, \mathbf{X}, Z) = H_1(A, \mathbf{X}, Z) + H_2(A, \mathbf{X}, Z) + H_3(A, \mathbf{X}, Z).$$

*It suffices to show that I, III, IV, and V are $o_P(1)$.*

*Consider term I. Note that*

$$\frac{K_h(A-a)(Z - \tilde{m}_a^k(\mathbf{X}))}{\hat{p}^k(a|\mathbf{X})} + \tilde{m}_a^k(\mathbf{X}) - \frac{K_h(A-a)(Z - m_a(\mathbf{X}))}{p(a|\mathbf{X})} - m_a(\mathbf{X})$$
$$=H_1(A, \mathbf{X}, Z) + H_2(A, \mathbf{X}, Z) + H_3(A, \mathbf{X}, Z) = H(A, \mathbf{X}, Z).$$

*We then compute $\mathbb{E}[\|I\|^2] = \mathbb{E}[\|\sqrt{h}(\mathbb{P}_{N_k} - \mathbb{E})H\|^2]$. Indeed, we can decompose it into the sum of two terms as follows:*

$$\mathbb{E}[\|\sqrt{h}(\mathbb{P}_{N_k} - \mathbb{E})H\|^2] = \frac{h}{N_k^2}\mathbb{E}\left[\|\sum_{i\in\mathcal{D}_k}\{H(A_i, \mathbf{X}_i, Z_i) - \mathbb{E}[H(A_i, \mathbf{X}_i, Z_i)]\}\|^2\right]$$

$$=\underbrace{\frac{h}{N_k^2}\sum_{i\in\mathcal{D}_k}\mathbb{E}\left[\|H(A_i, \mathbf{X}_i, Z_i) - \mathbb{E}[H(A_i, \mathbf{X}_i, Z_i)]\|^2\right]}_{I_1}$$

$$+\underbrace{\frac{h}{N_k^2}\sum_{\substack{i,j\in\mathcal{D}_k\\i\neq j}}\mathbb{E}\left[\langle H(A_i, \mathbf{X}_i, Z_i) - \mathbb{E}[H(A_i, \mathbf{X}_i, Z_i)], H(A_j, \mathbf{X}_j, Z_j) - \mathbb{E}[H(A_j, \mathbf{X}_j, Z_j)]\rangle\right]}_{I_2}$$

*We can bound $I_1$. Since $I_1 = H - \mathbb{E}[H] = H_1 - \mathbb{E}[H_1] + H_2 - \mathbb{E}[H_2] + H_3 - \mathbb{E}[H_3]$, we have*

$$I_1 \lesssim \frac{h}{N_k^2}\sum_{p=1}^{3}\sum_{i\in\mathcal{D}_k}\mathbb{E}\left[\|H_p(A_i, \mathbf{X}_i, Z_i) - \mathbb{E}[H_p(A_i, \mathbf{X}_i, Z_i)]\|^2\right]$$

$$\lesssim \underbrace{\frac{h}{N_k^2}\sum_{i\in\mathcal{D}_k}\mathbb{E}[\|H_1(A_i, \mathbf{X}_i, Z_i)\|^2]}_{I_{1-1}} + \underbrace{\frac{h}{N_k^2}\sum_{i\in\mathcal{D}_k}\mathbb{E}[\|H_2(A_i, \mathbf{X}_i, Z_i)\|^2]}_{I_{1-2}} + \underbrace{\frac{h}{N_k^2}\sum_{i\in\mathcal{D}_k}\mathbb{E}[\|H_3(A_i, \mathbf{X}_i, Z_i)\|^2]}_{I_{1-3}}.$$

Note that $I_{1-1} = \frac{h}{N_k} \mathbb{E}\big[\|H_1(A, \mathbf{X}, Z)\|^2\big]$. We turn to consider $h\mathbb{E}\big[\|H_1(A, X, Z)\|^2\big]$. We thus have

$$
h\mathbb{E}[\|H_1(A, \mathbf{X}, Z)\|^2] = h\mathbb{E}\left[K_h(A-a)^2 \left\| \frac{Z\{p(a|\mathbf{X}) - \hat{p}^k(a|\mathbf{X})\}}{\hat{p}^k(a|\mathbf{X})p(a|\mathbf{X})} \right\|^2\right]
$$
$$
\leq ch\mathbb{E}[|p(a|\mathbf{x}) - \hat{p}^k(a|\mathbf{X})|^2 \mathbb{E}[K_h(A-a)^2 \|Z\|^2 |\mathbf{X}]].
$$

Although $Z = \mathcal{Y}^{-1}$ is a function, $\|Z\|$ is a scalar. Hence, $\mathbb{E}\big[\|Z\|^2 \mid A = a, \mathbf{X}\big]$ can be treated as a function of $a$. Hence, we can express

$$
\mathbb{E}\big[\|Z\|^2 \mid A = a + uh, \mathbf{X}\big]
$$
$$
= \mathbb{E}\big[\|Z\|^2 \mid A = a, \mathbf{X}\big] + \partial_a \mathbb{E}\big[\|Z\|^2 \mid A = a, \mathbf{X}\big]uh + \frac{\partial_{aa}^2 \mathbb{E}\big[\|Z\|^2 \mid A = a, \mathbf{X}\big]u^2h^2}{2} + O_P(h^3).
$$

Further, since

$$
p(a + uh|\mathbf{X}) = p(a|\mathbf{x}) + \partial_a p(a|\mathbf{X})uh + \frac{\partial_{aa}^2 p(a|\mathbf{X})u^2h^2}{2} + O(h^3),
$$

we have

$$
\mathbb{E}\left[K_h(A-a)^2 \|Z\|^2 |\mathbf{X}\right] = \int \mathbb{E}\left[K_h(A-a)^2 \|Z\|^2 |A = s, \mathbf{X}\right]p(s|\mathbf{X})ds
$$
$$
= \frac{1}{h} \int \mathbb{E}\left[K(u)^2 \|Z\|^2 \mid A = a + uh, \mathbf{X}\right]p(a + uh|\mathbf{X})du
$$
$$
= \frac{1}{h}\left(\int K(u)^2 du\right)\mathbb{E}\big[\|Z\|^2 \mid A = a, \mathbf{X}\big]p(a|\mathbf{X})
$$
$$
+ \frac{h^2}{h}\left(\int K(u)^2 u^2 \, du\right)\mathbb{E}\big[\|Z\|^2 \mid A = a, \mathbf{X}\big]\frac{\partial_{aa}^2 p(a|\mathbf{X})}{2}
$$
$$
+ \frac{h^2}{h}\left(\int K(u)^2 u^2 du\right)\partial_a \mathbb{E}\big[\|Z\|^2 \mid A = a, \mathbf{X}\big]\partial_a p(a|\mathbf{X})
$$
$$
+ \frac{h^2}{h}\left(\int K(u)^2 u^2 \, du\right)\frac{\partial_{aa}^2 \mathbb{E}\big[\|Z\|^2 \mid A = a, \mathbf{X}\big]}{2}p(a|\mathbf{X}) + O_P(h^2).
$$

Hence, we have

$$
h\mathbb{E}\big[\|H_1(A, \mathbf{X}, Z)\|^2\big] \lesssim h\mathbb{E}\left[\left|p(a|\mathbf{X}) - \hat{p}^k(a|\mathbf{X})\right|^2 \mathbb{E}\big[K_h(A-a)^2 \|Z\|^2 |\mathbf{X}\big]\right]
$$
$$
= \left(\int K(u)^2 du\right)\underbrace{\mathbb{E}\left[\left|p(a|\mathbf{X}) - \hat{p}^k(a|\mathbf{X})\right|^2 \mathbb{E}\big[\|Z\|^2 |A = a, \mathbf{X}\big]p(a|\mathbf{X})\right]}_{I_{1-1a}}
$$
$$
+ h^2\left(\int K(u)^2 u^2 \, du\right)\underbrace{\mathbb{E}\left[\left|p(a|\mathbf{x}) - \hat{p}^k(a|\mathbf{X})\right|^2 \mathbb{E}\big[\|Z\|^2 |A = a, \mathbf{X}\big]\frac{\partial_{aa}^2 p(a|\mathbf{X})}{2}\right]}_{I_{1-1b}}
$$
$$
+ h^2\left(\int K(u)^2 u^2 du\right)\underbrace{\mathbb{E}\left[\left|p(a|\mathbf{x}) - \hat{p}^k(a|\mathbf{X})\right|^2 \partial_a \mathbb{E}\big[\|Z\|^2 |A = a, \mathbf{X}\big]\partial_a p(a|\mathbf{X})\right]}_{I_{1-1c}}
$$
$$
+ h^2\left(\int K(u)^2 u^2 \, du\right)\underbrace{\mathbb{E}\left[\left|p(a|\mathbf{x}) - \hat{p}^k(a|\mathbf{X})\right|^2 \frac{\partial_{aa}^2 \mathbb{E}\big[\|Z\|^2 |A = a, \mathbf{X}\big]}{2}p(a|\mathbf{X})\right]}_{I_{1-1d}} + O(h^3).
$$

We find the bounds of $I_{1-1a}$, $I_{1-1b}$, $I_{1-1c}$, and $I_{1-1d}$. Note that, according to the given conditions, we have

$$
I_{1-1a}, \, I_{1-1b}, \, I_{1-1c}, \, I_{1-1d} \lesssim \mathbb{E}[|p(a|\mathbf{X}) - \hat{p}^k(a|\mathbf{X})|^2] \leq (\mathbb{E}[|p(a|\mathbf{X}) - \hat{p}^k(a|\mathbf{X})|^4])^{\frac{1}{2}} \leq \rho_p^2.
$$

As a result, we conclude that

$$
I_{1-1} \lesssim \mathbb{E}[|p(a|\mathbf{X}) - \hat{p}^k(a|\mathbf{X})|^2] + O(h^3) \leq \rho_p^2 + O(h^3).
$$

*We therefore have*

$$I_{1-1} = O(\frac{1}{N_k}\rho_p^2 + \frac{h^2}{N_k}\rho_p^2 + O(h^3)).$$

*To bound $I_{1-2} = \frac{h}{N_k}\mathbb{E}\big[\|H_2(A, \mathbf{X}, Z)\|^2\big]$, we consider $h\mathbb{E}\big[\|H_2(A, \mathbf{X}, Z)\|^2\big]$. Indeed, we have*

$$h\mathbb{E}\big[\|H_2(A, \mathbf{X}, Z)\|^2\big] = h\mathbb{E}\left[K_h(A - a)^2\left\|\frac{\hat{p}^k(a|\mathbf{X})m_a(\mathbf{X}) - p(a|\mathbf{X})\tilde{m}_a^k(\mathbf{X})}{\hat{p}^k(a|\mathbf{X})p(a|\mathbf{X})}\right\|^2\right]$$

$$\leq ch\mathbb{E}\big[\left\|\hat{p}^k(a|\mathbf{X})m_a(\mathbf{X}) - p(a|\mathbf{X})m_a(\mathbf{X})\right\|^2\mathbb{E}[K_h(A - a)^2|\mathbf{X}]\big]$$

$$+ ch\mathbb{E}\big[\left\|p(a|\mathbf{X})m_a(\mathbf{X}) - p(a|\mathbf{X})\tilde{m}_a^k(\mathbf{X})\right\|^2\mathbb{E}[K_h(A - a)^2|\mathbf{X}]\big].$$

*We simplify the quantity $\mathbb{E}[K_h(A - a)^2|\mathbf{X}]$. Standard derivations give*

$$\mathbb{E}\left[K_h(A - a)^2|\mathbf{X}\right] = \frac{\left(\int K(u)^2 du\right)p(a|\mathbf{X})}{h} + \frac{\left(\int u^2 K(u)^2 du\right)\partial_{aa}^2 p(a|\mathbf{X})h}{2} + O_P(h^2).$$

*As a result, we have*

$$h\mathbb{E}\big[\|H_2(A, \mathbf{X}, Z)\|^2\big] \leq c\mathbb{E}\left[\left\|\hat{p}^k(a|\mathbf{X})m_a(\mathbf{X}) - p(a|\mathbf{X})m_a(\mathbf{X})\right\|^2\left(\int K(u)^2 du\right)p(a|\mathbf{X})\right]$$

$$+ ch^2\mathbb{E}\left[\left\|\hat{p}^k(a|\mathbf{X})m_a(\mathbf{X}) - p(a|\mathbf{X})m_a(\mathbf{X})\right\|^2\frac{\left(\int u^2 K(u)^2 du\right)\partial_{aa}^2 p(a|\mathbf{X})}{2}\right]$$

$$+ c\mathbb{E}\left[\left\|p(a|\mathbf{X})m_a(\mathbf{X}) - p(a|\mathbf{X})\tilde{m}_a^k(\mathbf{X})\right\|^2\left(\int K(u)^2 du\right)p(a|\mathbf{X})\right]$$

$$+ ch^2\mathbb{E}\left[\left\|p(a|\mathbf{x})m_a(\mathbf{X}) - p(a|\mathbf{X})\tilde{m}_a^k(\mathbf{X})\right\|^2\frac{\left(\int u^2 K(u)^2 du\right)\partial_{aa}^2 p(a|\mathbf{X})}{2}\right] + O(h^3)$$

*Therefore, we have*

$$I_{1-2} \lesssim \frac{1 + h^2}{N_k}\mathbb{E}\big[|\hat{p}^k(a|\mathbf{X}) - p(a|\mathbf{X})|^2\big] + \frac{1 + h^2}{N_k}\mathbb{E}\big[\|m_a(\mathbf{X}) - \tilde{m}_a^k(\mathbf{X})\|^2\big] + O(h^3)$$

$$\leq \frac{1 + h^2}{N_k}(\mathbb{E}\big[|\hat{p}^k(a|\mathbf{X}) - p(a|\mathbf{X})|^4\big])^{\frac{1}{2}} + \frac{1 + h^2}{N_k}(\mathbb{E}[\|m_a(\mathbf{X}) - \tilde{m}_a^k(\mathbf{X})\|^4])^{\frac{1}{2}} + O(h^3).$$

*Thus, we have*

$$I_{1-2} = O(\frac{1 + h^2}{N_k}\rho_p^2 + \frac{1 + h^2}{N_k}\rho_m^2 + h^3).$$

*To bound $I_{1-3}$, since $h\mathbb{E}\big[\|H_3(A, \mathbf{X}, Z)\|^2\big] \lesssim h\mathbb{E}\big[\|\tilde{m}_a^k(\mathbf{X}) - m_a(\mathbf{X})\|^2\big]$, we have*

$$I_{1-3} \lesssim \frac{h}{N_k}\mathbb{E}\big[\left\|\tilde{m}_a^k(\mathbf{X}) - m_a(\mathbf{X})\right\|^2\big] \leq \frac{h}{N_k}\rho_m^2.$$

*Thus, we have*

$$I_{1-3} = O(\frac{h}{N_k}\rho_m^2).$$

*Next, we bound $I_2$. Define*

$$G(A, \mathbf{X}, Z) := \frac{K_h(A - a)\{Z - \tilde{m}_a^k(\mathbf{X})\}}{\hat{p}^k(a|\mathbf{X})} + \tilde{m}_a^k(\mathbf{X}) - m_a(\mathbf{X})$$

$$F(A, \mathbf{X}, Z) := -\frac{K_h(A - a)\{Z - m_a(\mathbf{X})\}}{p(a|\mathbf{X})}.$$

*From the definitions of $G(A, \mathbf{X}, Z)$ and $F(A, \mathbf{X}, Z)$, we have $H(A, \mathbf{X}, Z) = G(A, \mathbf{X}, Z) + F(A, \mathbf{X}, Z)$. As a result, we have*

$$\big|\mathbb{E}\langle H(A_i, \mathbf{X}_i, Z_i) - \mathbb{E}\big[H(A_i, \mathbf{X}_i, Z_i)\big], H(A_j, \mathbf{X}_j, Z_j) - \mathbb{E}\big[H(A_j, \mathbf{X}_j, Z_j)\big]\rangle\big|$$

$$= \big|\mathbb{E}\langle G(A_i, \mathbf{X}_i, Z_i), G(A_j, \mathbf{X}_j, Z_j)\rangle - \langle\mathbb{E}\big[G(A_i, \mathbf{X}_i, Z_i)\big], \mathbb{E}\big[G(A_j, \mathbf{X}_j, Z_j)\big]\rangle\big|$$

$$+ \big|\mathbb{E}\langle G(A_i, \mathbf{X}_i, Z_i), F(A_j, \mathbf{X}_j, Z_j)\rangle - \langle\mathbb{E}\big[G(A_i, \mathbf{X}_i, Z_i)\big], \mathbb{E}\big[F(A_j, \mathbf{X}_j, Z_j)\big]\rangle\big|$$

$$+ \big|\mathbb{E}\langle G(A_j, \mathbf{X}_j, Z_j), F(A_i, X_i, Z_i)\rangle - \langle\mathbb{E}\big[G(A_j, \mathbf{X}_j, Z_j)\big], \mathbb{E}\big[F(A_i, \mathbf{X}_i, Z_i)\big]\rangle\big|$$

$$+ \big|\mathbb{E}\langle F(A_i, \mathbf{X}_i, Z_i), F(A_j, \mathbf{X}_j, Z_j)\rangle - \langle\mathbb{E}\big[F(A_i, \mathbf{X}_i, Z_i)\big], \mathbb{E}\big[F(A_j, \mathbf{X}_j, Z_j)\big]\rangle\big|.$$

*Consider* $\left|\mathbb{E}\langle G(A_i, \mathbf{X}_i, Z_i), G(A_j, \mathbf{X}_j, Z_j)\rangle - \langle\mathbb{E}\big[G(A_i, \mathbf{X}_i, Z_i)\big], \mathbb{E}\big[G(A_j, \mathbf{X}_j, Z_j)\big]\rangle\right|$. *We have*

$$\left|\mathbb{E}\langle G(A_i, \mathbf{X}_i, Z_i), G(A_j, \mathbf{X}_j, Z_j)\rangle - \langle\mathbb{E}\big[G(A_i, \mathbf{X}_i, Z_i)\big], \mathbb{E}\big[G(A_j, \mathbf{X}_j, Z_j)\big]\rangle\right|$$
$$\leq|\mathbb{E}\langle G(A_i, \mathbf{X}_i, Z_i), G(A_j, \mathbf{X}_j, Z_j)\rangle| + |\langle\mathbb{E}\big[G(A_i, \mathbf{X}_i, Z_i)\big], \mathbb{E}\big[G(A_j, \mathbf{X}_j, Z_j)\big]\rangle|$$
$$\overset{\diamond}{\leq}\|\mathbb{E}\big[G(A_i, \mathbf{X}_i, Z_i)\big]\|\|\mathbb{E}\big[G(A_j, \mathbf{X}_j, Z_j)\big]\| + \|\mathbb{E}\big[G(A_i, \mathbf{X}_i, Z_i)\big]\|\|\mathbb{E}\big[G(A_j, \mathbf{X}_j, Z_j)\big]\|$$
$$=2\|\mathbb{E}\big[G(A, \mathbf{X}, Z)\big]\|^2.$$

$\overset{\diamond}{=}$ *holds due to the fact that* $(A_i, \mathbf{X}_i, Z_i)$ *and* $(A_j, \mathbf{X}_j, Z_j)$ *are independent of each other and the Cauchy Schwartz inequality. Similarly, we have*

$$\left|\mathbb{E}\langle G(A_i, \mathbf{X}_i, Z_i), F(A_j, \mathbf{X}_j, Z_j)\rangle - \langle\mathbb{E}\big[G(A_i, \mathbf{X}_i, Z_i)\big], \mathbb{E}\big[F(A_j, \mathbf{X}_j, Z_j)\big]\rangle\right|$$
$$\leq2\|\mathbb{E}\big[G(A, \mathbf{X}, Z)\big]\|\|\mathbb{E}\big[F(A, \mathbf{X}, Z)\big]\|,$$
$$\left|\mathbb{E}\langle F(A_i, \mathbf{X}_i, Z_i), G(A_j, \mathbf{X}_j, Z_j)\rangle - \langle\mathbb{E}\big[F(A_i, \mathbf{X}_i, Z_i)\big], \mathbb{E}\big[G(A_j, \mathbf{X}_j, Z_j)\big]\rangle\right|$$
$$\leq2\|\mathbb{E}\big[F(A, \mathbf{X}, Z)\big]\|\|\mathbb{E}\big[G(A, \mathbf{X}, Z)\big]\|,$$

*and*

$$\left|\mathbb{E}\langle F(A_i, \mathbf{X}_i, Z_i), F(A_j, \mathbf{X}_j, Z_j)\rangle - \langle\mathbb{E}\big[F(A_i, \mathbf{X}_i, Z_i)\big], \mathbb{E}\big[F(A_j, \mathbf{X}_j, Z_j)\big]\rangle\right|$$
$$\leq2\|\mathbb{E}\big[F(A, \mathbf{X}, Z)\big]\|^2.$$

*Thus, we have*

$$\left|\mathbb{E}\langle H(A_i, \mathbf{X}_i, Z_i) - \mathbb{E}\big[H(A_i, \mathbf{X}_i, Z_i)\big], H(A_j, \mathbf{X}_j, Z_j) - \mathbb{E}\big[H(A_j, \mathbf{X}_j, Z_j)\big]\rangle\right|$$
$$\leq2\|\mathbb{E}\big[G(A, \mathbf{X}, Z)\big]\|^2 + 4\|\mathbb{E}\big[F(A, \mathbf{X}, Z)\big]\|\|\mathbb{E}\big[G(A, \mathbf{X}, Z)\big]\| + 2\|\mathbb{E}\big[F(A, \mathbf{X}, Z)\big]\|^2$$
$$=2\big(\|\mathbb{E}\big[G(A, \mathbf{X}, Z)\big]\| + \|\mathbb{E}\big[F(A, \mathbf{X}, Z)\big]\|\big)^2 \lesssim \|\mathbb{E}\big[G(A, \mathbf{X}, Z)\big]\|^2 + \|\mathbb{E}\big[F(A, \mathbf{X}, Z)\big]\|^2.$$

*Note that* $\|\mathbb{E}[G(A, \mathbf{X}, Z)]\| = \|\mathbb{E}[\mathbb{E}[G(A, \mathbf{X}, Z)|\mathbf{X}]]\| \leq \mathbb{E}[\|\mathbb{E}[G(A, \mathbf{X}, Z)|\mathbf{X}]\|]$, *we have*

$$\|\mathbb{E}[G(A, \mathbf{X}, Z)]\|^2 \leq (\mathbb{E}[\|\mathbb{E}[G(A, \mathbf{X}, Z)|\mathbf{X}]\|])^2 \leq \mathbb{E}[\|\mathbb{E}[G(A, \mathbf{X}, Z)|\mathbf{X}]\|^2].$$

*Thus, it suffices to consider* $\|\mathbb{E}\big[G(A, \mathbf{X}, Z)|\mathbf{X}\big]\|$ *and* $\|\mathbb{E}\big[F(A, \mathbf{X}, Z)|\mathbf{X}\big]\|$. *Now, from the definition of* $G(A, \mathbf{X}, Z)$, *we have*

$$\mathbb{E}\big[G(A, \mathbf{X}, Z)|\mathbf{X}\big] = \frac{(m_a(\mathbf{X}) - \tilde{m}_a^k(\mathbf{X}))(p(a|\mathbf{X}) - \hat{p}^k(a|\mathbf{X}))}{\hat{p}^k(a|\mathbf{X})}$$
$$+ \frac{(m_a(\mathbf{X}) - \tilde{m}_a^k(\mathbf{X}))(\int u^2 K(u)du)\partial_{aa}^2 p(a|\mathbf{X})h^2}{2\hat{p}^k(a|\mathbf{X})}$$
$$+ \frac{\partial_a\mathbb{E}[Z|A = a, \mathbf{X}](\int u^2 K(u)du)\partial_a p(a|\mathbf{X})h^2}{\hat{p}^k(a|\mathbf{X})}$$
$$+ \frac{p(a|\mathbf{x})(\int u^2 K(u)du)\partial_{aa}^2\mathbb{E}[Z|A = a, \mathbf{X}]h^2}{2\hat{p}^k(a|\mathbf{X})} + O_P(h^3).$$

*Thus, we have*

$$\|\mathbb{E}\big[G(A, \mathbf{X}, Z)|\mathbf{X}\big]\|$$
$$\lesssim\|(m_a(\mathbf{X}) - \tilde{m}_a^k(\mathbf{X}))\|\|(p(a|\mathbf{X}) - \hat{p}^k(a|\mathbf{X}))\| + \|m_a(\mathbf{X}) - \tilde{m}_a^k(\mathbf{X})\|\|\partial_{aa}^2 p(a|\mathbf{X})|h^2$$
$$+ \|\partial_a\mathbb{E}[Z|A = a, \mathbf{X}]\|\|\partial_a p(a|\mathbf{X})|h^2 + |p(a|\mathbf{X})|\|\partial_{aa}^2\mathbb{E}[Z|A = a, \mathbf{X}]\|h^2 + O_P(h^3).$$

*Similarly, we have*

$$\mathbb{E}\big[F(A, \mathbf{X}, Z)|\mathbf{X}\big] = -\frac{\partial_a\mathbb{E}[Z|A = a, \mathbf{X}](\int u^2 K(u)du)\partial_a p(a|\mathbf{X})h^2}{p(a|\mathbf{X})}$$
$$- \frac{p(a|\mathbf{X})(\int u^2 K(u)du)\partial_{aa}^2\mathbb{E}[Z|A = a, \mathbf{X}]h^2}{2p(a|\mathbf{X})} + O_P(h^3)$$

*and*

$$\|\mathbb{E}\big[F(A, \mathbf{X}, Z)|\mathbf{X}\big]\| \lesssim \|\partial_a\mathbb{E}[Z|A = a, \mathbf{X}]\|\|\partial_a p(a|\mathbf{X})|h^2 + |p(a|\mathbf{X})|\|\partial_{aa}^2\mathbb{E}[Z|A = a, \mathbf{X}]\|h^2 + O_P(h^3).$$

*Then, we compute* $\mathbb{E}[\|\mathbb{E}\big[G(A, \mathbf{X}, Z)|\mathbf{X}\big]\|^2]$ *and* $\mathbb{E}[\|\mathbb{E}\big[F(A, \mathbf{X}, Z)|\mathbf{X}\big]\|^2]$. *Note that*

$$\|\mathbb{E}\big[G(A, \mathbf{X}, Z)|\mathbf{X}\big]\|^2 \lesssim \|(m_a(\mathbf{X}) - \tilde{m}_a^k(\mathbf{X}))\|^2|(p(a|\mathbf{X}) - \hat{p}^k(a|\mathbf{X}))|^2 + O_P(h^4)$$
$$\Rightarrow \mathbb{E}[\|\mathbb{E}\big[G(A, \mathbf{X}, Z)|\mathbf{X}\big]\|^2] \lesssim \big(\mathbb{E}[\|(m_a^\lambda(\mathbf{X}) - \tilde{m}_a^k(\mathbf{X}))\|^4]\big)^{\frac{1}{2}}\big(\mathbb{E}[|p(a|\mathbf{X}) - \hat{p}^k(a|\mathbf{X})|^4]\big)^{\frac{1}{2}} + O(h^4)$$

*and*
$$\|\mathbb{E}\big[F(A,\mathbf{X},Z)|\mathbf{X}\big]\|^2 \lesssim O_P(h^4) \Rightarrow \mathbb{E}[\|\mathbb{E}\big[F(A,\mathbf{X},Z)|\mathbf{X}\big]\|^2] \lesssim O(h^4).$$

*Hence, we conclude that $I_2 = O(h\rho_p^2\rho_m^2 + h^5)$. Combining all the results, we can conclude that $I = o_P(1)$.*

*Consider $\|III\|$. We have*

$$\|III\| \leq \left\|\sqrt{h}\mathbb{E}_{N_k}\left[\frac{\{\tilde{m}_a^k(\mathbf{X}) - m_a(\mathbf{X})\}\{\hat{p}^k(a|\mathbf{X}) - K_h(A-a)\}}{\hat{p}^k(a|\mathbf{X})}\right]\right\|$$

$$+ \left\|\sqrt{h}\mathbb{E}_{N_k}\left[K_h(A-a)(Z - m_a(\mathbf{X}))\frac{(p(a|\mathbf{X}) - \hat{p}^k(a|\mathbf{X}))}{\hat{p}^k(a\mid\mathbf{X})p(a\mid\mathbf{X})}\right]\right\|$$

$$\lesssim \sqrt{h}\left\|\mathbb{E}_{N_k}\left[\frac{\{\tilde{m}_a^k(\mathbf{X}) - m_a(\mathbf{X})\}}{\hat{p}^k(a|\mathbf{X})}\mathbb{E}_{N_k}[\{\hat{p}^k(a|\mathbf{X}) - K_h(A-a)\}|\mathbf{X}]\right]\right\|$$

$$+ \sqrt{h}\left(\mathbb{E}_{N_k}\big[\,\|(Z - m_a(\mathbf{X}))\|^2\,\big]\right)^{\frac{1}{2}}\left(\mathbb{E}_{N_k}\big[|p(a|\mathbf{X}) - \hat{p}^k(a\mid\mathbf{X})^2|\big]\right)^{\frac{1}{2}}$$

$$\lesssim \sqrt{h}\left\|\mathbb{E}_{N_k}\left[\{\tilde{m}_a^k(\mathbf{X}) - m_a(\mathbf{X})\}\{\hat{p}^k(a|\mathbf{X}) - p(a|\mathbf{X}) - \frac{h^2}{2}\partial_{aa}^2 p(a|\mathbf{X})\int u^2 K(u)du + O(h^3)\}\right]\right\| + \sqrt{h}\rho_p$$

$$\lesssim \sqrt{h}\mathbb{E}_{N_k}\left[\left\|\{\tilde{m}_a^k(\mathbf{X}) - m_a(\mathbf{X})\}\{\hat{p}^k(a|\mathbf{X}) - p(a|\mathbf{X})\}\right\|\right]$$

$$+ \sqrt{h}\mathbb{E}_{N_k}\left[\frac{h^2}{2}\left\|\{\tilde{m}_a^k(\mathbf{X}) - m_a(\mathbf{X})\}\partial_{aa}^2 p(a|\mathbf{X})\int u^2 K(u)du\right\|\right] + O(h^{\frac{7}{2}}) + \sqrt{h}\rho_p$$

$$\lesssim \sqrt{h}\left(\mathbb{E}_{N_k}\big[\,\left\|\tilde{m}_a^k(\mathbf{X}) - m_a(\mathbf{X})\right\|^2\,\big]\right)^{\frac{1}{2}}\left(\mathbb{E}_{N_k}\big[\,|\,\hat{p}^k(a|\mathbf{X}) - p(a|\mathbf{X})\,|^2\,\big]\right)^{\frac{1}{2}}$$

$$+ \sqrt{h}\frac{h^2\left(\int u^2 K(u)du\right)}{2}\left(\mathbb{E}_{N_k}\big[\,\left\|\tilde{m}_a^k(\mathbf{X}) - m_a(\mathbf{X})\right\|^2\,\big]\right)^{\frac{1}{2}}\left(\mathbb{E}_{N_k}\big[|\partial_{aa}^2 p(a|\mathbf{X})|^2\big]\right)^{\frac{1}{2}} + O(h^{\frac{7}{2}}) + \sqrt{h}\rho_p.$$

*We can therefore conclude that $III = O(h^{\frac{1}{2}}\rho_p + h^{\frac{1}{2}}\rho_p\rho_m + h^{\frac{3}{2}}\rho_m + h^{\frac{7}{2}})$, and hence $III = o_P(1)$.*

*Consider the term IV. Note that*

$$\|IV\|^2 = \underbrace{\frac{1}{N_k^2}\sum_{i\in\mathcal{D}_k}\left\|\{1 - \frac{K_h(A_i - a)}{\hat{p}^k(a|\mathbf{X}_i)}\}\{D_a^k(\mathbf{X}_i)\}\right\|^2}_{IV_1}$$

$$+ \underbrace{\frac{1}{n_k^2}\sum_{\substack{i,j\in\mathcal{D}_k \\ i\neq j}}\langle\{1 - \frac{K_h(A_i - a)}{\hat{p}^k(a|\mathbf{X}_i)}\}\{D_a^k(\mathbf{X}_i)\}, \{1 - \frac{K_h(A_j - a)}{\hat{p}^k(a|\mathbf{X}_j)}\}\{D_{a,k}(\mathbf{X}_j)\}\rangle}_{IV_2}$$

*It can be shown that $IV_1 \lesssim \frac{1}{N_k}\sum_{i\in\mathcal{D}_k}\left\|D_a^k(\mathbf{X}_i)\right\|^2$. Besides, we can show that*

$$\|\hat{m}_a^k - \tilde{m}_a^k\|^2 = \frac{1}{N_k}\mathbb{E}\left[\sum_{i\in\mathcal{D}_k}\left\|D_a^k(\mathbf{X}_i)\right\|^2\right].$$

*Now, for any $\delta > 0$, using Markov inequality gives*

$$\mathbb{P}\left\{\frac{1}{N_k}\sum_{i\in\mathcal{D}_k}\left\|D_a^k(\mathbf{X}_i)\right\|^2 \geq \delta^{-1}\|\hat{m}_a^k - \tilde{m}_a^k\|^2\right\} \leq \delta\frac{\frac{1}{N_k}\mathbb{E}\big[\sum_{i\in\mathcal{D}_k}\left\|D_a^k(\mathbf{X}_i)\right\|^2\big]}{\|\hat{m}_a^k - \tilde{m}_a^k\|^2} = \delta.$$

*Under the Convergence Assumptions, we conclude that*
$$IV_1 = O_P(\|\hat{m}_a^k - \tilde{m}_a^k\|^2) = O_P(N^{-2} + N^{-1}\nu_N^2 + N^{-1}\alpha_N^2).$$

*For the quantity $IV_2$, we notice that*

$$IV_2 \leq \frac{1}{N_k^2}\sum_{\substack{i,j\in\mathcal{D}_k \\ i\neq j}}\left\|\{1 - \frac{K_h(A_i - a)}{\hat{p}^k(a|\mathbf{X}_i)}\}\{D_a^k(\mathbf{X}_i)\}\right\|\left\|\{1 - \frac{K_h(A_j - a)}{\hat{p}^k(a|\mathbf{X}_j)}\}\{D_a^k(\mathbf{X}_j)\}\right\|$$

$$\leq \frac{N_k - 1}{N_k}\frac{1}{N_k}\sum_{i\in\mathcal{D}_k}\left\|\{1 - \frac{K_h(A_i - a)}{\hat{p}^k(a|\mathbf{X}_i)}\}\{D_a^k(\mathbf{X}_i)\}\right\|^2 \leq \frac{1}{N_k}\sum_{i\in\mathcal{D}_k}\left\|\{1 - \frac{K_h(A_i - a)}{\hat{p}^k(a|\mathbf{X}_i)}\}\{D_a^k(\mathbf{X}_i)\}\right\|^2.$$

Similarly, we can show that $IV_2 = O_P(N^{-2} + N^{-1}\nu_N^2 + N^{-1}\alpha_N^2)$. Hence, $IV = O_P(N^{-1} + N^{-\frac{1}{2}}\nu_N + N^{-\frac{1}{2}}\alpha_N)$ which implies that $IV = o_P(1)$.

Consider the term V. Write

$$\mathbb{P}_{N_k}\left[\frac{K_h(A-a)R}{\hat{p}^k(a|\mathbf{X})}\right] = \mathbb{P}_{N_k}\left[\frac{K_h(A-a)R}{p(a|\mathbf{X})}\right] + \mathbb{P}_{N_k}\left[\frac{K_h(A-a)R}{\hat{p}^k(a|\mathbf{X})} - \frac{K_h(A-a)R}{p(a|\mathbf{X})}\right].$$

The second term is dominated by the first term since the second term involves the difference between the estimated density function $\hat{p}^k(a|\mathbf{X})$ and the true density function $p(a|\mathbf{X})$. Now, we consider the first term and we have

$$\mathbb{E}\left[\frac{1}{N_k}\sum_{i=1}^{N_k}\left\|\frac{K_h(A_i-a)R_i}{p(a|\mathbf{X}_i)}\right\|\right] \leq \frac{c}{N_k}\sum_{i=1}^{N_k}\mathbb{E}[\mathbb{E}[K_h(A_i-a)|\mathbf{X}_i]\|R_i\|]$$

$$=c\left\{\frac{1}{N_k}\sum_{i=1}^{N_k}\mathbb{E}[p(a|\mathbf{X}_i)\|R_i\|] + \frac{h^2\int u^2 K(u)du}{2}\frac{1}{N_k}\sum_{i=1}^{N_k}\mathbb{E}[\partial_{aa}^2 p(a|\mathbf{X}_i)\|R_i\|]\right\} + O(h^3)$$

$$\lesssim (1+h^2)\left(\mathbb{E}\left[\frac{1}{N_k}\sum_{i=1}^{N_k}\|R_i\|^2\right]\right)^{\frac{1}{2}} + O(h^3)$$

Using Lemma 2 and assumptions on $\alpha_N$ and $\nu_N$, we have $V = O_P((1+h^2)(\alpha_N+\nu_N)+h^3)$ which implies that $V = o_P(1)$. As a result, we have

$$\sqrt{Nh}(\hat{\triangle}_{a;DML}^{C;h} - \triangle_a)$$

$$=\sqrt{Nh}\left(\frac{1}{N}(N_1\hat{\psi}_{a,1} + N_2\hat{\psi}_{a,2}) - \psi_a\right) + \sqrt{Nh}\mathbb{E}\left[\frac{K_h(A-a)(\mathcal{L}\mathcal{Y} - m_a(\mathbf{X}))}{p(a|\mathbf{X})}\right]$$

$$=\sqrt{Nh}\left[(\mathbb{P}_N - \mathbb{E}_N)\{\varphi(A,\mathbf{X},\mathcal{Y})\} + \mathbb{E}_N\left[\frac{K_h(A-a)(\mathcal{L}\mathcal{Y} - m_a(\mathbf{X}))}{p(a|\mathbf{X})}\right]\right] + o_P(1)$$

$$=\sqrt{Nh}\left[\mathbb{P}_N\{\varphi(A,\mathbf{X},\mathcal{Y})\} - \triangle_a\right] + o_P(1).$$

Besides, we can rewrite the above equality as follows:

$$\sqrt{Nh}\left\{\hat{\triangle}_{a;DML}^{C;h} - \triangle_a - \mathbb{E}\left[\frac{K_h(A-a)(\mathcal{Y}^{-1} - m_a(\mathbf{X}))}{p(a|\mathbf{X})}\right]\right\} = \sqrt{Nh}\left[(\mathbb{P}_N - \mathbb{E})\{\varphi(A,\mathbf{X},\mathcal{Y})\}\right] + o_P(1).$$

Now, note that

$$\mathbb{E}\left[\frac{K_h(A-a)(\mathcal{Y}^{-1} - m_a(\mathbf{X}))}{p(a|\mathbf{X})}\right] = \mathbb{E}\left[\frac{1}{p(a|\mathbf{X})}\mathbb{E}\left[K_h(A-a)(\mathcal{Y}^{-1} - m_a(\mathbf{X}))|\mathbf{X}\right]\right]. \tag{21}$$

Detailed derivations show that give Eqn. equation 21 equals the following quantity:

$$h^2\left(\int u^2 K(u)du\right)\underbrace{\left\{\mathbb{E}\left[\partial_a\mathbb{E}[\mathcal{Y}^{-1}|\mathbf{X}, A=a]\frac{\partial_a p(a|\mathbf{X})}{p(a|\mathbf{X})}\right] + \mathbb{E}\left[\frac{\partial_{aa}^2\mathbb{E}[\mathcal{Y}^{-1}|\mathbf{X}, A=a]}{2}\right]\right\}}_{B_a} + O(h^3).$$

Finally, by the Central Limit Theorem, $\sqrt{Nh}\left[(\mathbb{P}_N - \mathbb{E})\{\varphi(A,\mathbf{X},\mathcal{Y})\}\right]$ converges weakly to a Gaussian process. The proof is now completed.

## I    COMPLETE RESULTS OF SIMULATION EXPERIMENTS

Table 4: The numerical experiment results for the Dist-DR, Dist-IPW, and Dist-DML estimators on continuous treatment values $A = -0.05, 0.00, 0.05$. The reported values are averages across 100 experiments, with standard deviations indicated in parentheses. The best results are highlighted in bold.

| | Q=0.1 | Q=0.2 | Q=0.3 | Q=0.4 | Q=0.5 | Q=0.6 | Q=0.7 | Q=0.8 | Q=0.9 | Error |
|---|---|---|---|---|---|---|---|---|---|---|
| | | | | | $A = -0.05$ | | | | | |
| **Ground** | 0.0107 | 0.0440 | 0.1030 | 0.2161 | 0.4783 | 0.7404 | 0.8535 | 0.9125 | 0.9458 | |
| **DR** | 0.0089 | 0.0347 | 0.1344 | 0.2858 | 0.4666 | 0.6528 | 0.8129 | 0.9133 | 0.9207 | |
| | (0.0048) | (0.0027) | (0.0029) | (0.0044) | (0.0062) | (0.0077) | (0.0097) | (0.0119) | (0.0162) | |
| **DR-e** | 0.0018 | 0.0092 | 0.0314 | 0.0697 | 0.0116 | 0.0876 | 0.0406 | 0.0008 | 0.0252 | 0.0309 |
| **IPW** | 0.0060 | 0.0537 | 0.1211 | 0.2380 | 0.4702 | 0.6904 | 0.8008 | 0.8626 | 0.9117 | |
| | (0.0004) | (0.0015) | (0.0034) | (0.0067) | (0.0135) | (0.0205) | (0.0238) | (0.0255) | (0.0269) | |
| **IPW-e** | 0.0046 | 0.0097 | 0.0181 | 0.0219 | 0.0081 | 0.0500 | 0.0527 | 0.0499 | 0.0341 | 0.0277 |
| **DML** | 0.0064 | 0.0550 | 0.1269 | 0.2508 | 0.4913 | 0.7201 | 0.8376 | 0.9040 | 0.9533 | |
| | (0.0005) | (0.0009) | (0.0008) | (0.0020) | (0.0013) | (0.0021) | (0.0015) | (0.0019) | (0.0020) | |
| **DML-e** | 0.0042 | 0.0110 | 0.0239 | 0.0347 | 0.0131 | 0.0204 | 0.0159 | 0.0086 | 0.0075 | **0.0155** |
| | | | | | $A = 0.00$ | | | | | |
| **Ground** | 0.0112 | 0.0462 | 0.1083 | 0.2271 | 0.5026 | 0.7782 | 0.8970 | 0.9591 | 0.9941 | |
| **DR** | 0.0101 | 0.0364 | 0.1412 | 0.3009 | 0.4917 | 0.6879 | 0.8561 | 0.9609 | 0.9670 | |
| | (0.0050) | (0.0027) | (0.0029) | (0.0045) | (0.0064) | (0.0079) | (0.0100) | (0.0124) | (0.0169) | |
| **DR-e** | 0.0011 | 0.0099 | 0.0329 | 0.0738 | 0.0109 | 0.0903 | 0.0409 | 0.0019 | 0.0271 | 0.0321 |
| **IPW** | 0.0071 | 0.0557 | 0.1240 | 0.2424 | 0.4817 | 0.7064 | 0.8190 | 0.8809 | 0.9293 | |
| | (0.0004) | (0.0014) | (0.0031) | (0.0063) | (0.0129) | (0.0208) | (0.0240) | (0.0257) | (0.0271) | |
| **IPW-e** | 0.0041 | 0.0095 | 0.0158 | 0.0153 | 0.0210 | 0.0718 | 0.0780 | 0.0781 | 0.0648 | 0.0398 |
| **DML** | 0.0080 | 0.0589 | 0.1353 | 0.2658 | 0.5195 | 0.7591 | 0.8846 | 0.9547 | 1.0039 | |
| | (0.0006) | (0.0010) | (0.0009) | (0.0021) | (0.0034) | (0.0024) | (0.0019) | (0.0021) | (0.0019) | |
| **DML-e** | 0.0032 | 0.0127 | 0.0270 | 0.0387 | 0.0169 | 0.0190 | 0.0124 | 0.0044 | 0.0098 | **0.0160** |
| | | | | | $A = 0.05$ | | | | | |
| **Ground** | 0.0118 | 0.0486 | 0.1138 | 0.2387 | 0.5283 | 0.8179 | 0.9428 | 1.0080 | 1.0448 | |
| **DR** | 0.0114 | 0.0381 | 0.1483 | 0.3167 | 0.5179 | 0.7246 | 0.9014 | 1.0109 | 1.0158 | |
| | (0.0053) | (0.0028) | (0.0030) | (0.0046) | (0.0065) | (0.0082) | (0.0104) | (0.0129) | (0.0176) | |
| **DR-e** | 0.0004 | 0.0105 | 0.0345 | 0.0780 | 0.0103 | 0.0933 | 0.0414 | 0.0029 | 0.0290 | 0.0334 |
| **IPW** | 0.0095 | 0.0619 | 0.1338 | 0.2637 | 0.5153 | 0.7570 | 0.8781 | 0.9443 | 0.9954 | |
| | (0.0005) | (0.0016) | (0.0034) | (0.0069) | (0.0136) | (0.0206) | (0.0238) | (0.0254) | (0.0267) | |
| **IPW-e** | 0.0023 | 0.0133 | 0.0200 | 0.0251 | 0.0130 | 0.0609 | 0.0646 | 0.0637 | 0.0493 | 0.0347 |
| **DML** | 0.0104 | 0.0649 | 0.1434 | 0.2831 | 0.5460 | 0.7994 | 0.9307 | 1.0034 | 1.0558 | |
| | (0.0005) | (0.0009) | (0.0006) | (0.0017) | (0.0010) | (0.0023) | (0.0013) | (0.0014) | (0.0017) | |
| **DML-e** | 0.0013 | 0.0164 | 0.0296 | 0.0444 | 0.0178 | 0.0184 | 0.0121 | 0.0046 | 0.0110 | **0.0173** |

## J    EMPIRICAL DATASETS

### J.1    NHANES DATASET AND PREPROCESSING

The National Health and Nutrition Examination Survey (NHANES) is a program designed to evaluate the health and nutritional status of adults and children in the United States. The survey collected information mainly via interviews and physical examination. The data about participants' demographics, diet, socioeconomics, and health status are gathered from interviews. The physical examination includes medical, dental, physiological assessments, and laboratory tests executed by medical professionals. Data from this survey can be used to explore the effects of risk factors on diseases and physical activity patterns. In this section, we study the causal effect of the employment status (full-time, part-time, or unemployed) on physical activity levels in American adults, with data obtained from the NHANES 2005-2006[1].

In the experiment, the causal effect of working hours on physical activity levels is our interest, so we also need to measure physical activity levels. Note that, in the 2005-2006 cycle of NHANES, participants ages 6 and older were required to wear an Actigraph uniaxial accelerometer on a waist belt throughout all non-sleeping hours for seven days. The accelerometer-based devices recorded vertical acceleration for successive 1-minute intervals and can measure the physical activity intensity. According to the recorded data, the physical activity intensity ranged from 0 to 32767 cpm.

To obtain robust and reliable results, we applied the following preprocessing procedures.

Firstly, the activity intensity data which were questionable were excluded. Then, we only considered subjects with activity intensity records for at least 10 hours a day and valid records for 4 days at least. Thirdly, we only took into account the observations with intensity values between 1 and 1200 cpm since most intensity values lie in the interval in this data set. The objective is to measure the distribution. We delete samples with smaller than 100 observations.

Besides employment status, we also utilized some covariates, including gender, race/ethnicity, educational level, marital status, occupation code as a categoric variable, age, BMI, the total number of people in the family, and the poverty income ratio (PIR) as continuous variables. After removing samples with missing data, there were 2762 participants left.

### J.2    STATISTICAL ANALYSIS

Table 5 displays the statistical analysis, which was stratified by working hours, i.e., Full-time stands for working hours exceeds 35 hours in one week, Part-time stands for working hours between 1 and 35 hours in one week, and Unemployed stands for working hours equal 0 hour in one weak. We offer the means and standard deviations of the five continuous variables. As shown in Table 5, the full-time group has an average age of 41.09 years old, with the lowest standard deviation (12.80). The part-time job has the biggest average age and standard deviation with values of 42.18 and 17.26, respectively, indicating that the age range of part-time work is relatively large. The highest average income and PIR are found among those who work full-time, which is consistent with our intuition. For the number of people living together, there is no significant difference between the average of the full-time group and part-group, whereas the unemployed have a relatively high value. The average BMIs in the three groups do not show much difference.

In terms of categoric variables, we show their counts and percentages in each category. The number of males and females is slightly imbalance in the full-time and part-time groups. However, in the unemployed group, the number of men and women is highly unbalanced. $78.41\%$ are female, and only $21.59\%$ are male. For race/ethnicity, the percentages of various races in the full-time and part-time groups are similar. Non-Hispanic White, Non-Hispanic Black, and Mexican American are the first three races in the two groups. Different from the two groups, Mexican American in the unemployed makes up the highest proportion, followed by non-Hispanic White. The proportions of different education levels vary in three groups. The people with the highest educational level account for the highest proportion of the full-time group and the lowest proportion of the unemployed group, whereas the results of the lowest education level are the opposite. Among the three groups, more than half are married. Never married is the second largest proportion.

---

[1]https://wwwn.cdc.gov/nchs/nhanes/ContinuousNhanes/Default.aspx?BeginYear=2005.

Table 5: Statistics Analysis of Covariates

| Covariates | Full-Time (N=1759) | Part-Time (N=438) | Unemployed (N=565) |
|---|---|---|---|
| **Age** | 41.09±12.80 | 42.18±17.26 | 39.58±16.62 |
| **Poverty Income Ratio** | 3.15±1.56 | 2.78±1.62 | 1.86±1.40 |
| **BMI** | 28.78±6.17 | 28.05±6.24 | 28.88±7.22 |
| **Total number of people in the Household** | 3.26±1.58 | 3.17±1.51 | 3.84±1.74 |
| **Gender** | | | |
| Male | 1000(56.85%) | 182(41.55%) | 122(21.59%) |
| Female | 759(43.15%) | 256(58.45%) | 443(78.41%) |
| **Race/Ethnicity** | | | |
| Mexican American | 371(21.09%) | 69(15.75%) | 197(34.87%) |
| Other Hispanic | 63(3.58%) | 17(3.88%) | 28(4.96%) |
| Non-Hispanic White | 843(47.92%) | 232(52.97%) | 192(33.98%) |
| Non-Hispanic Black | 409(23.25%) | 99(22.60%) | 121(21.42%) |
| Other Race (including multi-racial) | 73(4.15%) | 21(4.79%) | 27(4.78%) |
| **Educational level** | | | |
| Less Than 9th Grade | 151(8.58%) | 35(7.99%) | 107(18.94%) |
| 9-11th Grade | 206(11.71%) | 47(10.73%) | 127(22.48%) |
| High School Grad/GED | 387(22.00%) | 104(23.74%) | 129(22.83%) |
| Some College/ AA degree | 553(31.44%) | 166(37.90%) | 139(24.60%) |
| College Graduate or above | 462(26.26%) | 86(19.63%) | 63(11.15%) |
| **Marital status** | | | |
| Married | 1008(57.31%) | 263(60.05%) | 308(54.51%) |
| Widowed | 29(1.65%) | 14(3.20%) | 36(6.37%) |
| Divorced | 172(9.78%) | 35(7.99%) | 30(5.31%) |
| Separated | 53(3.01%) | 9(2.05%) | 16(2.83%) |
| Never married | 314(17.85%) | 87(19.96%) | 107(18.94%) |
| Living with partner | 183(10.40%) | 30(6.85%) | 68(12.04%) |

## K    COMPUTATION INFRASTRUCTURE

All experiments are run on Dell 7920 with Intel(R) Xeon(R) Gold 6250 CPU at 3.90GHz, and a set of NVIDIA Quadro RTX 6000 GP. All models are implemented in Python 3.8. The versions of the main packages of our code are: Pytorch 1.8.1+cu102, Sklearn: 0.23.2, Numpy: 1.19.2, Pandas: 1.1.3, Matplotlib: 3.3.2.

