# OpenReview forum: "Causal Inference on Distributional Outcomes under Continuous Treatments"
_ICLR.cc/2024/Conference — ICLR 2024 Conference Withdrawn Submission_

### Official Review · Reviewer_5HfR · 2023-10-26

**Soundness:** 2 fair
**Presentation:** 2 fair
**Contribution:** 1 poor
**Rating:** 5
**Confidence:** 3

**Summary:**

This paper addresses the need for causal inference methods that can handle the distributional nature of response variables. It introduces a novel framework for causal inference in a vector space that utilizes the Wasserstein metric. This paper introduces three estimators: Dist-DR, Dist-IPW, and Dist-DML. The paper thoroughly explores the statistical properties of these estimators and validates their practical utility through two experiments, demonstrating their efficacy in handling real-world scenarios involving distributional responses.

**Strengths:**

- The paper exhibits clear organization, with well-defined formulations throughout.
- The problem is quite novel and interesting.

**Weaknesses:**

The problem this paper aims to address is very clear, which is to extend previous cases where the outcome was scalar or vector to include distributions. The motivation behind this is quite straightforward. However, I am unsure if there are real-world examples that support the problem constructed in this paper, and this is my primary concern. If not, then this is essentially a 'toy paper'.

The theory and methods of using p-Wasserstein metric to measure the distance between two different distributions are quite well-developed. This paper directly incorporates causal effect estimation and p-Wasserstein metric together to construct this new task.

**Questions:**

- For continuous treatments, do we need to define the causal effect map, which is similar to the Average Treatment Effect (ATE), or is it sufficient to calculate just a similar estimate as the Average Dose-Response Function (ADRF) as we did previously?
- In addition to the p-Wasserstein metric, it is better to provide more metrics to measure the differences between two distributions. Since this is the first paper studying distributional outcomes, more methods are needed to provide support.
- Have you attempted to integrate popular neural network-based methods, such as VCNet and Transformer methods, into your approach?

---

### Official Review · Reviewer_nqXb · 2023-10-30

**Soundness:** 3 good
**Presentation:** 2 fair
**Contribution:** 2 fair
**Rating:** 3
**Confidence:** 3

**Summary:**

The authors study a model where A is a continuous treatment, X are unrestricted covariates, and Y(a) is a distribution-valued potential outcome. The authors propose a generalization of the dose response for this setting, and analyze a doubly robust estimator with kernel smoothing for the dose.

**Strengths:**

1. The paper’s contribution appears to be in its definition of an estimand for this generalized setting and its proposal of a doubly robust estimator for it, generalizing the dose response analysis of previous work.

2. Overall, the results are presented in a logical manner. Further justification of the estimand would help to make the case for this paper’s significance.

3. The results seem right at a high level, though I did not check carefully. I wonder whether the conditions for Theorem 1 are too strong.

**Weaknesses:**

1. The basic data setting and estimand were not clearly explained. Figure 6 in the appendix should perhaps go in the main text. I have clarifying questions below.

2. For a paper that studies a generalization of dose response curves with a doubly robust estimator, I would expect more references to the mature literature on doubly robust estimation of dose response curves, e.g “Nonparametric methods for doubly robust estimation of continuous treatment effects” and the references within “Doubly debiased machine learning nonparametric inference with continuous treatments”.

3. Some of the framing needs to be improved. “There arises a need for causal inference methods that can account for the distributional nature of responses,…” This framing suggests that no previous methods can accommodate estimation of treatment effects with continuous treatments and functional outcomes. This isn’t true. For example, RKHS methods have been proposed that allow the space of Y to be quite general and for the space of A to be continuous. See e.g. the extensions in “Kernel methods for causal functions: Dose, heterogeneous, and incremental response curves” where Y is an element of a Polish space.

4. Some statements were imprecise. “However the estimated relation from the observed dataset can be biased since the dataset is not always randomized.” This is not the source of bias. Averaging over covariates X takes care of this.

I will improve my score if these aspects are addressed.

**Questions:**

1. First some basic clarifications:

1a. Is the setting one where, for each observation, we get to see X, A, and samples from the distribution Y?

1b. Definition 2 was hard to follow. Please write explicitly the integral in the expectation. It seems we are taking a random function, finding its best expected approximation in the Wasserstein sense (where the expectation is over the randomness of the function), then inverting this now deterministic function. is that right?

1c. Proposition 1 seems to take a random function, invert it, then take its mean to obtain a deterministic function. Is that right? Again please write out the integrals.

Now some substantial questions:

2. Is there any previous work that studies the causal effect map so defined? Why is this a good definition?

3. The rate conditions in Theorem 1 are in terms of the rho objects, which are like L4 rates rather than L2 rates. This is at odds with the previous doubly robust dose response literature, e.g. those cited above, though possibly similar to “Orthogonal statistical learning”. Please explain why this stronger condition is necessary. Is it unavoidable?

I will improve my score if these aspects are addressed.

---

### Official Review · Reviewer_F8tv · 2023-11-01

**Soundness:** 2 fair
**Presentation:** 2 fair
**Contribution:** 2 fair
**Rating:** 3
**Confidence:** 3

**Summary:**

In this paper, the authors study the problem of causal effect estimation, where the outcomes for each unit are represented as continuous values. The causal estimates discussed in prior literature often consider the outcomes for each unit to be represented as scalar or vector values. However, for many practical settings, these responses can be distributions. These distributions are a specialized case of functional outcomes and are closely associated with the field of functional data analysis (FDA). While recent works have attempted to study the causal impact of treatments on functional outcomes, they often operate within spaces that might compromise the inherent random structure of the distributional outcome. Building upon previous research, this paper introduces a new non-parametric framework and three distinct cross-fitted estimators for inferring causal effects when the treatment variable assumes continuous values. The study delves into the asymptotic properties of these estimators, providing crucial insights into their statistical performance and reliability. Additionally, empirical validation through two experiments corroborates the proposed estimator's consistency with theoretical findings, marking the threefold contributions of this research.

**Strengths:**

— The paper studies an important research question and provides robust estimators for a causal effect estimand under the continuous treatment regime.

— The paper is well-written, along with empirical evidence for robustness on synthetic datasets.

**Weaknesses:**

— The importance of the new estimand, causal effect map, particularly the difference of the inverse of potential outcomes, remains unclear at present. It must be compared to elucidate why it is considered an effective causal quantity as opposed to solely working with the difference of potential outcomes.

— Restating the causal assumptions concerning the inverses is necessary as they are not explicitly stated.

— The novelty is debatable, as the results and the related work suggest that the authors might have directly applied techniques from previous works to develop robust estimators for the new causal effect estimand.

— The experimental results lack benchmark datasets such as Lalonde, Twins, etc., which diminishes the comprehensive assessment and validation of the proposed estimators.

**Questions:**

— It would be interesting to compare with Lin et al. (2021) work more rigorously as this paper is a direct extension (as claimed in related work).

— \Delta_a(.) is not a function in Sec 4.1. Please use the notation carefully.

— Previously, Y(a) was a scalar; however, in Sec 4 and later, it is a distribution. Please indicate this in the related work.

— Sec 5 is not very useful, particularly without any explanations/interpretations regarding the theoretical result statements or their assumptions.

— The choice of synthetic data in Eq (14a) and (14b) has to be explained carefully; it seems arbitrarily chosen.

— In Sec 6, Pg 8: “These results are in line with our theoretical analysis” — Please explain this.

---

### Official Review · Reviewer_Hjb6 · 2023-11-06

**Soundness:** 3 good
**Presentation:** 3 good
**Contribution:** 2 fair
**Rating:** 5
**Confidence:** 3

**Summary:**

The paper introduces a new setting of treatment effect estimation, where for each unit of measurement the full distribution is observed as an outcome. This setting is different from traditional treatment effect estimation, where the main estimand is a scalar/vector (ATE) or conditional expectation of a scalar/vector random variable (CATE). In this paper, on the other hand, the authors consider potential outcomes to be distributions or elements of the Wasserstein space of distributions, and, thus, the main estimand is taken as averaged potential Wasserstein barycentre, namely a causal map. The estimation theory for this estimand is then developed based on the adapted potential outcomes framework. The theory introduces three estimators, I.e., inverse-propensity weighted (IPW) estimator, direct regression  (DR) estimator, and double machine learning (DML) estimator. Then, it was shown, that combined with the cross-fitting procedure, those estimators are consistent under mild conditions. Additionally, the estimators are asymptotically normal, and the  DML estimator possesses a double-robustness property. The authors verify their theoretical results with a synthetic benchmark and a real-world case study.

**Strengths:**

I find the proposed setting pretty novel and original. The paper was clearly written, and much effort is dedicated to explaining the difference from traditional treatment effect estimation (e.g., I appreciate Figures 7 and 8). Also, the theoretic results are rigorous and extend the stream of double robust estimation literature.

**Weaknesses:**

I have three main concerns regarding the paper:
1. From a practical point of view in my opinion, there are no distinctions between two estimands, namely, the newly introduced “causal map” and a simple “distribution of the potential outcomes”. That is, in the context of the decision-making from observational data they both would have the same semantic meaning. Also, in the majority of the real-world applications, we could not distinguish between both, as we only have access to the samples, drawn from the distributions, but not the distributions themselves for each unit of measurement. Therefore, I wonder, why couldn’t we just convert the problem into the estimation of the density of the potential outcomes, e.g., [1]? Therein, each unit of observation actually yields several observations, which are simply organized in one long-format table.
2. Another issue with the proposed setting is that it can be seen as treatment effect estimation over time with a single-time intervention. Merely all the motivational examples in the paper described datasets, where for each unit of measurement we observed time-series data, i.e., non-i.i.d. data. Then, by inferring the distributions for each unit we drop the time ordering information, which is an obvious loss of information. I encourage authors to add a discussion on this issue.
3.  A lot of details on the implementation are missing in Sec. 6 or Appendix. For example, it is unclear, how the functional regression is implemented, so that one can ensure that the estimated function is a CDF. Also, no details on the hyperparameters search or the confidence intervals estimation are given. Another issue with the experiments is that the synthetic dataset generator (Eq. 14a) seems to yield values of CDFs, which are larger than 1 or smaller than 0, which is not possible by the definition.

Also, there are some minor points:
- Some related works are missing. For example, the approach of smoothing the Dirac delta function with kernels for IPW or DML estimators was employed in [2]. Also, the whole body of work on estimating distributions of potential outcomes is missing, e.g., [1, 3-5].
-  The presentation of the results in Table 1 can be enhanced. E.g., now it looks like Dist-DR-MAE or Dist-DML-MAE are distinct methods.

I am eager to increase my score if the authors address the above-mentioned issues.


References:
[1] Kennedy, E. H., Balakrishnan, S., & Wasserman, L. (2021). Semiparametric counterfactual density estimation. arXiv preprint arXiv:2102.12034.
[2] Kallus, N., & Zhou, A. (2018, March). Policy evaluation and optimization with continuous treatments. In International conference on artificial intelligence and statistics (pp. 1243-1251). PMLR.
[3] Melnychuk, V., Frauen, D., & Feuerriegel, S. (2023, July). Normalizing flows for interventional density estimation. In International Conference on Machine Learning (pp. 24361-24397). PMLR.
[4] Kallus, N., & Oprescu, M. (2023, April). Robust and agnostic learning of conditional distributional treatment effects. In International Conference on Artificial Intelligence and Statistics (pp. 6037-6060). PMLR.
[5] Martinez-Taboada, D., & Kennedy, E. H. (2023). Counterfactual Density Estimation using Kernel Stein Discrepancies. arXiv preprint arXiv:2309.16129.

**Questions:**

See questions in the "weaknesses".